

# Thermal infrared laser heterodyne spectro-radiometry for solar occultation atmospheric CO$_2$ measurements

Alex Hoffmann[1], Neil A. Macleod[1], Marko Huebner[1], Damien Weidmann[1]

[1]Space Science and Technology Department (RAL Space), STFC Rutherford Appleton Laboratory, Harwell Campus, Didcot, OX11 0QX, UK

*Correspondence to*: Damien Weidmann (damien.weidmann@stfc.ac.uk)

**Abstract.** This technology demonstration paper reports on the development, demonstration, performance assessment and initial data analysis of a benchtop prototype quantum cascade laser heterodyne spectro-radiometer, operating within a narrow spectral window of ~1 cm$^{-1}$ around 953.1 cm$^{-1}$ in transmission mode, and coupled to a passive Sun tracker. The instrument has been specifically designed for accurate dry air total column, and potentially vertical profile, measurements of CO$_2$. Data from over 8 months of operation in 2015 near Didcot, UK, confirm that atmospheric measurements with noise levels down to four times the shot noise limit can be achieved with the current instrument. Over the 8 month period, spectra with spectral resolutions of 60 MHz (0.002 cm$^{-1}$) and 600 MHz (0.02 cm$^{-1}$) have been acquired with median signal-to-noise ratios of 113 and 257, respectively, and a wavenumber calibration uncertainty of 0.0024 cm$^{-1}$.

Using the optimal estimation method and RFM as the radiative transfer forward model, prior analysis and theoretical benchmark modelling had been performed with an observation system simulator (OSS) to target an optimized spectral region of interest. The selected narrow spectral window includes both CO$_2$ and H$_2$O ro-vibrational transition lines to enable the measurement of dry air CO$_2$ column from a single spectrum. The OSS and preliminary retrieval results yield roughly 8 degrees of freedom for signal for an arbitrarily chosen a priori state with relatively high uncertainty. Preliminary total column mixing ratios obtained are consistent with GOSAT monthly data. At a spectral resolution of 60 MHz with an acquisition time of 90 s, instrumental noise propagation yields an error of around 1.5 ppm on the dry air total column of CO$_2$, exclusive of biases and geophysical parameters errors at this stage.

**Keywords:** Laser Heterodyne Radiometry; QCL; atmospheric CO$_2$; thermal infrared sounding; instrument development

# 1 Introduction

The accelerating rise of global levels of carbon dioxide in the atmosphere, most likely due to human activity and anthropogenic emissions (Le Quéré et al., 2013), are believed to be the main driver of (anthropogenic) climate change (Myhre et al., 2013). Within the global carbon biogeochemical cycle (Ciais et al., 2013), the regional quantification of terrestrial (and to a lesser extent, oceanic) sources, sinks and feedbacks remains uncertain, particularly in the tropics (e.g. (Le



Quéré et al., 2009; Schimel et al., 2015). Likewise, geographical patterns of $CO_2$ emissions from fossil fuel sources are needed to improve upon the regional land flux estimates (Peylin et al., 2013).

Known as the top-down approach, carbon exchange processes between the atmosphere and the underlying surface are frequently derived from variations in measured near-surface $CO_2$ concentrations through inverse techniques based on

atmospheric transport models, in order to identify the spatio-temporal distribution of fluxes e.g. (Gurney et al., 2002; Peylin et al., 2013). To achieve this aim, near-surface measurements (continuous or from flask samples, (Conway et al., 1994)) are commonly used. An increasingly dense and comprehensive observation network and monitoring system operating to high degrees of precision and accuracy is ever more important (Ciais et al., 2010; Marquis and Tans, 2008; Nisbet and Weiss, 2010), as the results of inversions can be very sensitive to the observational network of $CO_2$ concentrations (Peylin et al.,

2013). Requirements on measurement precision and accuracy are extremely demanding. For instance, the (WMO/GAW, 2014) expert group recommends a target measurement compatibility for the $CO_2$ in situ mole fraction (i.e. concentration or mixing ratio) in dry air ($X_{CO2}$) of 0.1/0.05 ppm in the Northern and Southern hemispheres, respectively, for well-mixed background air. For emission monitoring, an absolute accuracy of 5% or better of the increase with respect to the background is recommended.

The number and frequency of observations can be dramatically increased through remote sensing from space. This has been shown to be potentially very effective in reducing uncertainties related to carbon flux estimations (Boesch et al., 2011; Chevallier et al., 2009), although the assimilation of real data seems as yet not to provide the expected improvements (Chevallier et al., 2014; Houweling et al., 2015). Obviously, the signature of carbon exchange processes is strongest within the atmospheric planetary boundary layer (PBL). In summertime North America for example, persistent deficits of 4-20 ppm

were measured in the lowest 2-3 km of the atmosphere, attributed to a vegetation sink, with smaller deficits in the boundary layer or larger ones in the free troposphere hypothetically related to deep convective overturning (Gerbig et al., 2003). The largest continental variability of up to 20 ppm was found to be confined to the mixed and residual layers, typically limited in altitude between several hundreds of metres and roughly 2 km. Concentrations in the free troposphere were found to have less variability, typically up to ~3 ppm from a marine boundary layer reference. Furthermore, $CO_2$ increases from nocturnal

respiration are likely to be confined to the usually very shallow early-morning mixed layers.

Any remote sensing technique hoping to contribute to accurate $CO_2$ measurements hence needs to be sensitive to the concentration within this lowest atmospheric layer. Also, total column mixing ratios exhibit weaker signatures of surface exchange processes than do actual surface measurements, and were found to yield a less precise inversion than classical near-surface sampling (Chevallier et al., 2011). Consequently, requirements on uncertainty may even be tighter than for

surface instrumentation.

Already in 2001, (Rayner and O'Brien, 2001) specified total column monthly-averaged measurement precision requirements of 1.5-2.5 ppm for the utility of a (global) moderately sized network to compare with that of existing surface measurements in flux inversions, using simulations. The Total Carbon Column Observing Network (TCCON, (Wunch et al., 2011)), currently comprising over 20 operational sites equipped with high-resolution Fourier Transform Spectrometers, provides a



basis for present and future satellite observation calibration and validation (e.g. (Deng et al., 2014)). It sets a new benchmark of 0.2-0.25% (~1 ppm) total error for remotely sensed observations. (Deutscher et al., 2010) demonstrated 0.1% relative precision for one of the TCCON sites located in Darwin, Australia. Similar values for total column precision/accuracy over regional scales (around 1 ppm) are also set, targeted and quoted for spaceborne observations (Boesch et al., 2011; Butz et al., 2011; Ciais et al., 2010; Crisp et al., 2004; Kadygrov et al., 2009; Miller et al., 2007; Thompson et al., 2012), depending on observational frequency, often requiring significant efforts to improve retrievals and spectroscopy.

Besides the potential for global and frequent coverage from space, one possible advantage of column-integrated over surface measurements is that they may reduce uncertainties due to (mostly convective) vertical mixing and transport across the PBL inversion layer and throughout the free troposphere, and hence dilution of the signal (Rayner and O'Brien, 2001; Wunch et al., 2011; Yang et al., 2007), provided such dilution is efficient on the regional scales of interest.

Meanwhile, height-resolved vertical profiles of $CO_2$, in particular with data points within PBL and in the free troposphere above, constitute a valuable further resource for quantifying advective transport, constraining carbon flux estimates and verifying inversion results (Crevoisier et al., 2010; Gerbig et al., 2003; Stephens et al., 2007). They can also be used to evaluate the exchange between the PBL and the free troposphere above. Mostly, such data are acquired by profiling using sampling on-board aircraft, although differential absorption and integrated path lidar have also been deployed for both range-resolved and (mostly) path-mean active remote sensing of $CO_2$ in the PBL and free troposphere (Abshire et al., 2010; Gibert et al., 2008; Koch et al., 2008; Ramanathan et al., 2015).

Whilst TCCON currently provides the best benchmark in ground-based remote sensing of $CO_2$, the upfront investment in establishing a site and the subsequent running costs prevent the network from reaching a high density. In addition, given the size of the FTIR instrument used, the rapid deployment of ad-hoc temporary and configurable networks cannot be envisaged. With the objective to provide complementary instrumentation addressing these shortfalls, an alternative approach for generating high spectral resolution spectra of atmospheric transmittance is considered here.

This paper reports on the technical development of a lab- and ground-based thermal infrared (TIR) laser heterodyne spectro-radiometer (LHR) aiming at accurately characterizing total column and vertical profiles of $CO_2$ in solar occultation mode, building on past instruments successfully used for the passive remote sounding of a variety of trace gases in the atmosphere (Tsai et al., 2012; Weidmann et al., 2007a, 2007b, 2011a). The prototype instrument has been installed at the Rutherford Appleton Laboratory near Chilton (Didcot), UK, and operated in fair weather conditions since May 2015.

Theoretical principles and a historical perspective on laser heterodyne spectro-radiometry are given elsewhere (e.g. (Weidmann et al., 2007a) and references therein). In a nutshell, the broadband radiation from the scene, containing the spectral signatures of the gas(es) of interest, is mixed with laser light from a Local Oscillator (LO). In the simplest implementation, the LO frequency is continuously tunable and the tuning range determines the spectral coverage of the spectro-radiometer. The mixing amplifies the weak radiometric signal and spectrally down-converts it into the radio-frequency (RF) domain, where a high spectral resolution can be set with standard RF filters (the spectral response function of which can be precisely measured), and the signal is recorded with a square-law detector. LHR combines the unique



advantages of very high spectral resolution (resolving power $\sim 10^6$) and high sensitivity (ideally shot noise limited) with a narrow field-of-view (FOV). The use of single mode semiconductor lasers as LO, in this case a quantum cascade laser, restricts the spectral coverage of the LHR and narrow spectral window optimization is required to specifically targets one or a few trace gases (Weidmann et al., 2007a). High spectral resolution, combined with a high signal-to-noise ratio (SNR), is

useful to accurately measure an atmospheric lineshape, and thus to derive total column trace gas amounts to a high degree of precision, or alternatively, to deconvolve altitudinal (height-resolved) information, both immediately useful in the context of carbon cycle and anthropogenic emission studies. Compared with other remote sensing techniques, LHR also has a high potential for miniaturization and ruggedization, e.g. through hollow waveguide integration (Weidmann et al., 2011b), which makes it well suited for field, air- and spaceborne deployment, individually, or in a networked configuration.

The use of LHR for $CO_2$ monitoring is also currently being investigated by (Clarke et al., 2014; Melroy et al., 2015; Wilson et al., 2014). Their system operates in the short-wave IR (1.6 µm), leveraging telecoms distributed feedback lasers as the LO and associated photonics components to couple the solar radiation to a single-mode optical fibre connected to the solar trackers of the AERONET Sun photometer network. Conversely, we use Quantum Cascade Lasers (QCLs) to operate in the thermal infrared. Not only is a TIR LHR of interest to look into the measurement consistency across spectral bands, but the

ideal SNR of laser heterodyne systems scales with the wavelength. For instance, considering a radiation emitted from the Sun equivalent blackbody, the ideal LHR SNR is improved by $\sim$15 times at 960 cm$^{-1}$ compared to 6500 cm$^{-1}$. In addition, access to the fundamental intense ro-vibrational bands in the TIR is more suited for probing trace atmospheric species, and atmospheric scattering is smaller.

## 2 Prior analysis and benchmark performance modelling

### 2.1 Generic description of the approach

Distributed feedback semiconductor lasers exhibit narrow spectral windows of continuous frequency tuning. The central frequency of these narrow windows can be precisely tailored, either by material composition change for laser diodes, or band engineering for quantum cascade lasers. As far as atmospheric sounding is concerned, these features ought to be considered as part of prior analysis to define and select the optimized narrow spectral windows well suited for, in this particular case,

$CO_2$ remote sensing. This prior analysis may help the instrument design by a) maximizing the information content obtained by the measurements, b) considering the addition of alternate atmospheric constituents such as $H_2O$ to be measured simultaneously and hence allowing calculation of dry air $CO_2$ mixing ratios, and c) minimizing spectral interferences from other narrow band absorbers to limit the possibility of model errors. The general approach to prior analysis consists of building an observing system simulator (OSS) consisting of 1) a forward model describing the remote sensing scenario and

radiative transfer (instrument and atmosphere), 2) an optimum estimation retrieval algorithm, and 3) a set of analysis tools to quantify sounding performances.





The forward modelling component of the OSS built to analyse LHR performances includes the MIPAS Reference Forward Model, RFM (Dudhia, 1997), version 4.33. The program is iteratively called to calculate atmospheric radiative transfer and the associated Jacobian matrix ($\mathbf{K}$). RFM is set to use 100 internal layers (every 0.5 km up to 10 km, then every 1 km up to 90 km). Jacobians are calculated with applied perturbations centred on a retrieval level, and linearly tapered off up to the

adjacent retrieval levels. The HITRAN 2012 (Rothman et al., 2013) spectral database provides the molecular absorption line parameters; Voigt line shapes are used by RFM as default. For simulation purposes, pressure, temperature and potentially interfering species ($O_3$, $N_2O$, $NH_3$, $COF_2$) profiles have been set equal to the typical mid-latitude MIPAS 2001 reference atmosphere compiled for the MIPAS instrument on-board the ENVISAT satellite (Dudhia, 1997). Realistic, "true" reference $H_2O$ and $CO_2$ profiles are inserted that have been constructed to exhibit larger structural variability than the standard profiles.

A water vapour profile measured on 31 July 2015 by operational radiosonde has been converted from mass to volume mixing ratio before insertion. An artificial $CO_2$ profile has been generated with a depression in the PBL, hypothetically accounting for plant uptake of $CO_2$, and with random variability on a slowly varying background throughout the full atmospheric column, which is roughly aligned with in situ aircraft measurements used as reference by (Ramanathan et al., 2015).

The forward model also includes an LHR instrument model (Weidmann et al., 2007a), primarily to work out the SNR delivered by the instrument as function of instrumental parameters such as integration time, spectral resolution, sampling resolution, etc. With adequate detectors, laser heterodyne radiometry is ideally LO shot noise-limited (Weidmann et al., 2007a). In practice, real noise levels are expected to be above the shot noise limit. It is therefore realistic to simulate instrument performance at the shot noise limit to provide the ultimate benchmark reference against which experimental data

are to be compared. In the OSS, LO shot noise is mapped into the transmission domain, assuming the radiating background is an ideal blackbody at 5778 K (see Sect. 4.3). In other words, no account of potential broadband atmospheric absorption is factored in, hence simulating the upper limit in terms of performance. Random noise is added to the synthetic spectra assuming a normal distribution.

The second component of the OSS is the optimal estimation algorithm (OEA). Its role consists in providing the best estimate

possible of the quantities being measured given our knowledge of the atmospheric state and information provided by the LHR recorded spectra. $CO_2$ profiles and total column amounts are ultimately determined by an inverse modelling-based retrieval. This involves the non-linear least squares fitting of a forward-modelled spectrum ($F(x)$), integrating the trace gas concentrations to be estimated (i.e. the state vector $x$), to a measured spectrum ($y$) with measurement error ($\varepsilon$), expressed as an error covariance matrix $\mathbf{S}_\varepsilon$, and starting from a prior state ($x_a$) with an a priori covariance matrix $\mathbf{S}_a$. The details of the

OEA are beyond the scope of this paper. Suffice to say that the retrieval is built around a well-established methodology (Rodgers, 2000), using Bayesian statistics with Gaussian probability density functions, and a Levenberg-Marquardt iterative procedure to minimize the cost function ($\chi^2$):

$$\chi^2 = [y - F(x_i)]^T \mathbf{S}_\varepsilon^{-1} [y - F(x_i)] + [x_i - x_a]^T \mathbf{S}_a^{-1} [x_i - x_a], \tag{1}$$





Trace gases are typically (but not necessarily) solved for as the natural logarithm of their volume mixing ratio, to constrain the algorithm to positive concentrations and improve the handling of large dynamic ranges. The retrieval code can be run with the purely synthetic data provided by the forward model (pseudo-observations), thus completing the OSS to study the anticipated instrument performance.

In addition, the OEA offers quantitative tools to analyse the observing system performance and identify an optimal spectral region: the most useful of these are the Shannon information content ($H$) and the degrees of freedom for the signal ($d_s$) (Rodgers, 2000):

$$H = -0.5 \cdot \ln|\mathbf{I_n} - \mathbf{A}|, \tag{2}$$

$$d_s = tr(\mathbf{A}), \tag{3}$$

where $\mathbf{I_n}$ is the unit matrix of order $n$ (the number of state vector elements). The averaging kernel matrix $A_{ij} = \partial \hat{x}_i / \partial x_j$, giving the sensitivity of the retrieved state ($\hat{x}$) to the true state ($x$), is calculated as:

$$\mathbf{A} = (\mathbf{K^T S_\varepsilon^{-1} K + S_a^{-1}})^{-1} \mathbf{K^T S_\varepsilon^{-1} K}, \tag{4}$$

The combination of forward modelling, OEA, and the corresponding information content analysis tools completes the OSS and is then used in the practical case of $CO_2$ sounding using a LHR operating from the ground in solar occultation.

**2.2 Implementation for $CO_2$ sounding**

A preliminary survey of spectral regions aimed at assessing thermal infrared LHR performance for trace gas sounding, (Rose and Weidmann, 2013) identified the ($\nu_1 \leftarrow \nu_3$) hot absorption band centred around 960 cm$^{-1}$ as suitable for solar occultation measurements, unlike the much stronger $\nu_2$ fundamental band, centred at 662 cm$^{-1}$, which was better suited for nadir and limb emission soundings. At 960 cm$^{-1}$, the spectrum of $CO_2$ consists of a well-defined P-R band structure of absorption lines

typically spaced by ~2 cm$^{-1}$. In a first instance, the OSS was used iteratively in conjunction with a "sliding" narrow spectral window. Figure 1 shows the evolution of the information content for both $CO_2$ and $H_2O$ retrieval as a window of 1 cm$^{-1}$ is slid by 0.2 cm$^{-1}$ steps. $CO_2$ information content follows the ro-vibrational band structure. Focusing on the P branch, in which QCLs were more readily available at the time of the instrument development, the window centred at 953.1 cm$^{-1}$ was found to be a good compromise for optimum remote sounding of both $CO_2$ (22 bits) and $H_2O$ (20 bits). Additionally, in this window,

any other potential narrow-band atmospheric absorber contributes to less than 10$^{-3}$ in absorption based on a typical mid-latitude atmosphere. The $CO_2$ line retained for this work is the P10 transition around 952.880849 cm$^{-1}$ (10.5 μm) with an intensity of 1.9 × 10$^{-23}$ cm$^{-1}$/(molec.cm$^{-2}$). The $H_2O$ line within the narrow window is centred at 953.367430 cm$^{-1}$, has an intensity of 4.801 × 10$^{-24}$ cm$^{-1}$/(molec.cm$^{-2}$) and belongs to the $\nu_2$ fundamental band.

The corresponding averaging kernels (AKs) were calculated for illustration purposes using a dense vertical retrieval grid

with 0.5 km spacing and are shown in Fig. 1b and c. The AKs can be interpreted as the vertical sensitivity of the measurement, and show that the measurement should be most sensitive to the lowest tropospheric layers, in particular to $CO_2$ and $H_2O$ concentrations within the planetary boundary layer (PBL). Using the metric of the sum of the AKs < 0.8, the $CO_2$





sensitivity goes up to 25 km and then tails off at -0.05 km$^{-1}$, while the water sensitivity reaches 8 km and then drops at a rate of -0.5 km$^{-1}$. Since $CO_2$ is assumed to be a reasonably well-mixed gas (with the profile used for simulating the spectrum reflecting this), the averaging kernels decay with the atmospheric exponential pressure decrease with height, determining the absolute number of absorbing molecules within a given layer. Likewise, the averaging kernels for $H_2O$ decay with the typical

altitudinal distribution of water vapour, for which freeze drying results in only trace amounts above the tropopause.

Once the optimum LHR narrow window has been chosen, a more targeted analysis can establish the ideal benchmark against which measurements are to be compared. The simulation represents a best-case scenario of a shot-noise limited instrument without biases (systematic errors) or other error sources in the observing and retrieval system.

Within the OSS, the state vector $x$ concatenates the altitudinal profile of both $CO_2$ and $H_2O$ to be retrieved, as well as one

coefficient ($a_0$) that describes the linear mapping (radiance scaling) from atmospheric transmission to heterodyne signal and potential broadband absorption not included in the forward model. This coefficient is initially set to 1. The a priori $x_a$ is taken from the MIPAS reference atmosphere, standard errors on $x_a$ are arbitrarily set to 20%, 50% and 100% for $CO_2$, $H_2O$ and $a_0$, respectively, and $S_a$ is also assumed diagonal. These assumptions do not necessarily reflect the optimum conditioning of the retrieval; in the context of this work the objective is to benchmark the first outcomes from the instrumental development of

the TIR LHR and focus on instrument error. The altitudes on which to retrieve $CO_2$ and $H_2O$ have been chosen by iteration such that i) the size of the state vector $n$ is close to $d_s$, ii) the averaging kernels cover the entire retrievable column with approximate overlap at their half-maxima (see Fig. 1b and c), iii) the retrieved layers give enough vertical resolution to capture highest variability within the PBL, iv) retrieval sensitivity to the exact state vector configuration is limited, and v) the residuals after the fit exhibit no obvious bias. The solar elevation angle was set to 60 degrees. The simulated instrument

integration time was 50 ms, and ideal shot noise limited operation was assumed. The same instrument line shape (ILS) as measured for the real instrument, and similar values for the sampling resolution (0.001 cm$^{-1}$) have been introduced.

Using the OSS, an ensemble of 50 retrievals at 600 MHz (20 at 60 MHz) is run, each member differing only by the random seed used to generate the Gaussian measurement noise, used in turn to degrade the synthetic spectrum, and inserted (squared) onto the diagonal elements of $S_\varepsilon$. Results are shown in Fig. 2a and 2b in which the LHR spectral resolution was set to

60MHz (0.002 cm$^{-1}$) and 600 MHz (0.02 cm$^{-1}$) respectively. The upper part of the figures shows the actual LHR spectra overlaid with the fitted ones. Residuals from the fits are shown below. The lower panels show the retrieval outcomes. For each spectral resolution is shown from left to right: the $H_2O$ profile, the $CO_2$ profile, the $CO_2$ and $H_2O$ retrieval errors, and the averaging kernels. In the profile plots are shown: the a priori profile (red line), the "true" profile used in the OSS, the successive 50 retrieved ensemble member profiles (in transparent purple), and the first member profile from the set (black

with error bars). Both the retrieved water and $CO_2$ profiles (and in particular their ensemble average, not shown) converge very close to the "truth" used in the OSS. The variability of the 50 retrievals in the $CO_2$ case exhibits some unphysical oscillations, yet still within the one sigma error bars. Both measurement (M in the figure, from $S_M = G S_\varepsilon G^T$, where $G = \partial \hat{x}/\partial y$ is the gain matrix) and smoothing (S, from $S_s = (I_n - A) S_a (I_n - A)^T$) retrieval errors are shown. In the case of $CO_2$, retrieval error is dominated by measurement error, which implies that smoothing and the influence of the a priori on the



retrieved state are almost negligible. This is anticipated, since $x_a$ was specified with large uncertainties and without off-diagonal correlation terms on $S_a$. Conversely, the $H_2O$ error is dominated by smoothing error. This is particularly the case at higher altitudes, where the system is no longer very sensitive to the actual water vapour concentrations and a priori knowledge therefore significantly contributes to the solution. Note, though, that all the ensemble members are essentially

collocated, and that a (bias-free) LHR system seems to be very accurate in reconstituting the true, yet smoothed, $H_2O$ profile. From the retrieval profiles, column measurements can be obtained and are shown in Table 1. Loosely following e.g. (Boesch et al., 2011; O'Dell et al., 2012), the estimates of total vertical column amounts $VC$ and associated variances $\sigma_{VC}$ have been calculated, respectively, as:

$$VC_g = h_g^T \hat{x}_g, \tag{5}$$

$$\sigma_{VC_g}^2 = h_g^T \hat{S}_g h_g, \tag{6}$$

where the subscript $g$ indicates the vector section or block matrix corresponding to trace gas $g$ (either $CO_2$ or $H_2O$) of the retrieved state and error covariance matrix (identified by hat), $\hat{S} = (K^T S_\varepsilon^{-1} K + S_a^{-1})^{-1}$. To calculate the pressure weighting function $h_g$ the following steps are taken: i) the air's molecular number density profile is calculated from the ideal gas law using input temperature and pressure profiles interpolated onto a fine vertical grid up to the top-of-the-atmosphere, ii) for

each retrieval grid level $i$, the associated $h_{g,I}$ element is computed as the vertically integrated molecular number density between the mid-points (or ground level) centred on altitude $i$. and iii), the number of molecules of trace gas $g$ follows trivially through multiplication by the retrieved volume mixing ratio (VMR). The (moist air) total column-averaged VMR and error ($TC$ and $\sigma_{TC}$, Table 1) follow by dividing $VC$ ($\sigma_{VC}$) by the air's total column-integrated molecular number density.

An alternate error estimation has been carried out using the ensemble of 50 (20) retrievals from identical synthetic

observations and calculating the standard deviation over the TCs estimated for the individual ensemble members. $X_{CO2}$ is the column-averaged $CO_2$ mole fraction in dry air following the calculation in (Deutscher et al., 2010); its error has been estimated for illustrative purposes using the variance formula error propagation simplification with a surface pressure ($p_{srf}$) uncertainty arbitrarily set to 3hPa.

From Table 1, it appears that the errors calculated using error propagation (i.e. from $\hat{S}$) and the standard deviation calculated

over the ensemble (for total column amounts as for individual retrieval levels, not shown except in PBL), are consistent for $CO_2$, which gives confidence in the retrieval's robustness. Individually retrieved total column amounts are within <1 ppm of their true value and precision is down to <1 ppm. Individual profiles do not coincide with the true profile, and exhibit some of the oscillatory characteristic of under-constrained retrievals, but their ensemble average approaches the true profile. For $H_2O$, the retrieval error exceeds the standard deviation over the ensemble, which is likely due to smoothing. These results

imply that LHR can potentially comply with the high-accuracy requirements of $CO_2$ observing systems aimed at flux inversions, and can potentially discriminate the PBL from the free troposphere above, albeit with a relatively high uncertainty under the SNR condition of the OSS runs. The degrees of freedom for signal imply that 8 (or possibly more) independent pieces of information can be retrieved from a spectrum. The next sections investigate to what extent the



performance of a real system matches the predicted (shot noise-limited) one. Performances can be improved by using longer integration time. The conditions presented in the OSS correspond to only 90 seconds per measurement.

## 3 Instrument design, assembly and specification

### 3.1 Laser heterodyne receiver

The breadboard prototype $CO_2$ LHR assembled for this work follows the overall system design of previous iterations of the instrument (Weidmann et al., 2011a) with significant upgrades in terms of solar tracking, and a parallel mixing arrangement rather than convergent. The optical layout of the system is shown in Fig. 3a. The QCL operates single mode with an operating wavelength of ~10.47 μm (955 cm$^{-1}$) and a tuning range of ~7 cm$^{-1}$ via temperature and current modulation (Alpes Lasers SA). It operates in continuous mode at ambient temperatures (< 40°C) with up to 30 mW of power which is more

than sufficient for heterodyne detection. The object beam waist in the QCL facet plane was measured to be 8 μm. The diverging QCL beam is collimated by a moulded aspheric lens (component CL in Fig. 3.a; 4 mm effective focal length, 0.56 numerical aperture) and passed through a wire-grid polarizer used as an adjustable attenuator (ATT) to allow the optimum LO power to be selected. A beam expander follows, consisting of 2 off-axis parabolic mirrors (OAPM1 and OAPM2) in an anti-symmetrical arrangement to reduce potential aberrations. Upon reflection by the beam splitter (BS, wedged ZnSe, 25%

reflection), the beam is focused onto the cryogenically-cooled high-speed photodiode (PD1, 1GHz bandwidth HgCdTe photomixer, Raytheon) by OAPM3. Beam expansion is necessary to match the (image) beam waist to the detector element size (100 μm diameter). The requirement is to illuminate 80% of the photodiode active area with 99% of the LO beam encircled energy. Assuming a typical $M^2$ parameter of a QCL of 1.2, this yields a targeted beam waist of ~30 μm in the detector plane, and thus an overall magnification requirement of about three. The overall magnification of the LO optical

arm is 3.18.

The 75% of LO radiation transmitted by the BS is directed through a 76.10 ±0.01 mm long Germanium etalon for real-time relative laser frequency calibration during the QCL tuning. OAPM4 is used to focus the beam onto a slow TE-cooled photodiode (PD2, Vigo Systems PVM-4TE-10.6).

Additional optics channel the solar longwave radiation, collected by a solar tracker after transmission through the

atmosphere, onto the BS where it is spatially superimposed with the LO beam. The solar radiation goes through a longwave-pass filter (LWP), transmitting radiation above ~7.7 μm (90% transmission at 10.5 μm). The optical power reflected by the filter is directed onto a wide-area NIR broadband Germanium photodiode (NIR PD, New Focus model 2033, 3 dB spectral response from 1100-1600 nm) as a solar intensity diagnostic, alignment aid, and cloud monitor. The transmitted longwave radiation passes through an afocal system (OAPM4 and OAPM5) producing a Sun image at which a mechanical chopper is

positioned (MOD, model TTI C-995, operating at 1777 Hz) to provide signal amplitude referencing.



In order to estimate the heterodyne field of view and the aperture contributing to the heterodyne signal at the modulation plane (MOD), Gaussian beam back propagation of the LO image formed on PD1 is considered. At the MOD, the optical system provides a 6.4 magnification relative to the beam waist at the QCL facet plane. The back propagated LO beam divergence up to the final optic is ~0.52 mrad and represents the coherent FOV of the instrument. This corresponds to about

1/8 of the apparent solar disc angle.

No radiometric calibration is needed for the LHR operating in transmission mode. However, a calibrated source (a compact blackbody held at 1050°C) can be branched into the instrument's FOV via a flip mirror to quantitatively characterize the radiometric performance of the system.

The output of PD1 is fed in to an AC/DC splitter and an amplifier chain. The DC output is used to monitor the LO power

which is modulated as part of the frequency tuning. The AC component (heterodyne) is filtered through an RF bandpass filter, which defines the LHR spectral resolution and instrument lineshape. Filters with bandpass ranges of either 50-80 MHz, or 50-350 MHz have been used to set the double side band spectral resolution to 60 and 600 MHz respectively. The RF power is then detected by a Schottky diode (Eclipse Microwave EZR0120A3) whose output is fed into a lock-in amplifier (Ametek Signal Recovery 7265 DSP) synchronized with the chopper located at the MOD. The lock-in integration

time $\tau$ has been set to 50 ms for the majority of measurements. Under these conditions the total acquisition time per spectrum is 90 s.

Purpose-built LabVIEW software controls the signal acquisition process. A DAQ-card (NI USB-6259) acquires the in-phase and out-of-phase heterodyne signals from the lock-in amplifier, the DC signals from PD1, the calibration and reference signals from PD2 and NIR PD, and a collection of metadata, including operating parameters.

The QCL is controlled in temperature and current using low noise linear controllers. It has been housed in a custom-built high stability mount. Indeed, the sensitivity of the coherent mixing onto PD1 is critical and highly sensitive to the QCL facet position. Changes in laser temperature are used for coarse wavelength tuning, whilst a current ramp produces laser frequency sweeps over a ~1 cm$^{-1}$ span. To target the selected narrow spectral window identified, the QCL operates at 477 ± 85 mA and ~9.0°C.

**3.2 Passive solar tracker**

Previous LHR atmospheric measurement campaigns (Weidmann et al., 2007a, 2011a) were carried out using an active solar tracker using quadrant detectors and a feedback loop to maintain alignment with the Sun. This system was found to be extremely sensitive to cloud coverage, and systematically ceased normal operation upon the slightest reduction in collected solar brightness. This is not viable for the prospect of an unattended autonomous ground-based LHR operating in mid

latitudes. A benefit of TIR LHR is the provision of a significantly higher number of observations compared to alternate ground based sounders by leveraging on the fast measurement time and narrow FOV. In other words, to be able to pick up good quality data "between clouds", the system ought to be resilient to clouds passing by the FOV.



To enable this, a passive altazimuth tracking system has been developed that relies on a Sun trajectory algorithm. Our assembly consists of two protected gold-plated 50.8×25.4 mm$^2$ elliptically shaped flat mirrors on motorized rotation stages. The azimuth stage has a 50 mm diameter central aperture, through which the radiation is projected vertically downwards onto a folding mirror, passing through a BaF$_2$ window at the base of the assembly. The solar tracker is mounted in a remote-

and environment-controlled protective dome on the roof of a building with almost unobstructed hemispherical viewing capability. The LHR has been assembled on a portable workstation placed directly in front of the window port through which the solar radiation enters (Fig. 3b).

Since the LHR's FOV only covers a fraction of the apparent Sun disc, and assuming isotropic radiance as a first approximation, pointing accuracy and precision requirements are not stringent as far as the stability of solar power received

is concerned. However, very accurate and precise pointing knowledge is required in order to avoid significant bias at the data retrieval stage. Indeed, column retrieved data are highly sensitive to the effective air mass sounded, particularly at low elevation angles. (Gisi et al., 2011), for example, specify a 19 arcsec tracking precision requirement to achieve 0.1% trace gas total column precision at 10° elevation. Our motion system is built around two high-performance rotation stages (Thorlabs NR360S NanoRotators, controlled by a BSC102 APT stepper motor controller). These are specified to provide a

maximum accuracy over a 10° range of 5 arcmin (~0.08°) and a minimum incremental motion of <1 arcsec (~0.0002°). Mirror alignment, levelling, tracker systems testing, and pointing assessment have been performed using medium-distance (several metres) visible laser circuits and surveying equipment. Horizontally-level pointing during azimuth rotation over a range of 180° was achieved to within an error of ~0.2°. Repeated azimuth and elevation pointing precision from any angular position was measured to be < 1.8 arcmin (0.03°) without intermittent homing; it was found that homing was detrimental to

the precision.

A first iteration of the Sun trajectory algorithm embedded within the tracker's LabVIEW control software is the SUNAE algorithm by Michalsky, Harrison and Wiscombe, based on a revision of (Michalsky, 1988) without the atmospheric refraction correction. This specifies an accuracy of better than 0.1° for elevations above 9°, which can be improved to 0.01° with refraction correction, and was deemed one of the most accurate algorithms amongst those reviewed by (Blanco-Muriel

et al., 2001). This algorithm can be swapped with more accurate and complex ones if necessary (Blanc and Wald, 2012; e.g. Grena, 2012; Reda and Andreas, 2004), but this is unlikely to much improve the system if it is limited by the mechanical and alignment accuracy.

Solar tracking is performed in iterated discrete steps, typically 5-10 s apart. For an altazimuth mount, the apparent azimuth and elevation angular velocities are non-linear functions of date, time, and latitude, which we have calculated as numerical

derivatives. For our location, the azimuth angular velocity is larger than that for elevation and reaches a maximum at local noon around the summer solstice of 0.0081 °s$^{-1}$. With a 10 s update period, this corresponds to a 5 arcmin angular deviation or a trace over 16% of the Sun disc. Practically, this value is an upper limit and will usually be smaller. In the case of elevation, following the same reasoning, the worst case scenario pointing accuracy is 1.6 arcmin, which propagates to a <1%



relative accuracy on the column measurement for elevation >3°, assuming the air mass is inversely proportional to the sine of the elevation angle.

Experimentally, over the course of almost one year of operation, it was found that unsupervised tracking can satisfactorily be performed over several hours and up to a day without needing to recalibrate the tracker with no noticeable impact on the

measured signal. For comparison, the AERONET 4-quadrant active solar tracker used by (Wilson et al., 2014) has a stated accuracy of better than 0.1°. However, given the stringent accuracy requirements on $X_{CO2}$ translating into tolerances on air mass uncertainties, better pointing accuracy may eventually be required, especially at low elevation. This, in turn, is likely to be best achievable with a complementary active feedback mechanism. Pointing tolerances will need further evaluation using the full retrieval suite for error quantification, and proper accounting of measurement biases.

**4 Instrument characterization**

**4.1 Local oscillator**

In this implementation of the TIR LHR, a single fixed filter approach has been retained and the spectral components are obtained through the scanning of the local oscillator frequency. The QCL operating frequency is temperature- and current-dependent and was studied and measured to ensure control over the emitted laser frequency. The wavenumber $\sigma$ emitted as

function of current and temperature $T$ is modelled by the function given in Eq. (7). This equation describes the quadratic dependence of frequency on both current and temperature, as well as a cross-coupling term between the two tuning parameters. Fitting of the tuning data allows determining the parameters driving Eq. (7). These are given in Table 2 and are used to define the coarse operating point of the laser.

$$\sigma = \sigma_0 + i_1 \cdot I + i_2 \cdot I^2 + t_1 \cdot T + t_2 \cdot T^2 + x \cdot I \cdot T, \tag{7}$$

The high stability laser module allows operation from -15°C. The laser has a maximum current of 0.56 A, and a maximum operating temperature of 30°C. The laser threshold is 0.36 A at the lowest achievable temperature. Hence, the accessible tuning range is 950.78 to 955.37 cm$^{-1}$ and the laser can access the optimum narrow spectral window at about 953 cm$^{-1}$ determined in the previous section. An operating temperature of 9°C and a DC current of 0.477 A were chosen. Current tuning is performed using a sawtooth-waveform voltage ramp produced by a function generator that produces a current

modulation of ±85 mA.

The laser can deliver up to 20 mW of optical power, and attenuation is necessary to reach the LO power which optimizes the heterodyne SNR (~150 µW). Spectral characteristics were analysed using a Bristol wavemeter/spectrum analyser, and were found to be single mode throughout the operating range. Spatial profiles were checked to be TEM 00 ellipsoidal Gaussian (residual with Gaussian fit <5% of the peak intensity value), with additional signs of residual diffraction from the collimation

lens. The beam ellipticity is constantly ~10% and the M$^2$ parameter was evaluated to be ~1.2.





## 4.2 Frequency calibration

As a first approximation, ignoring contributions from potentially interfering lines, the total column is related to the area delimited by a single isolated spectral absorption line. Therefore, the accuracy of the retrieved signal is strongly dependant on the accuracy of the relative frequency calibration of the spectral data. To that end, real time frequency calibration is

conducted using the etalon described in the experimental section, which provides a free spectral range of 491.35 ± 0.07 MHz at 953 cm$^{-1}$ and 21°C.

The wavenumber axis associated with a spectral scan is determined by an identical frequency calibration procedure as described by (Tsai et al., 2012). A high-order polynomial is fitted to the resulting data from the etalon optical arm, giving the relative laser frequency calibration by steps of half a free spectral range (between two consecutive extrema in the etalon

trace). The relative wavenumber vector is shifted to absolute wavenumbers by matching the narrow $CO_2$ absorption line centre to the corresponding value tabulated in the HITRAN spectral database.

In order to gain insight into the stability of the relative frequency calibration throughout a full day, continuous spectral scans were recorded every 90 s to construct a long dataset of frequency calibration data (~05:00-19:00 UTC on 30 June 2015). The raw etalon signals over the full day are shown in Fig. 4a and frequency drifts are clearly perceptible throughout the 14 hour

measurement period. Frequency drifts are not an issue since they would not affect the relative frequency calibration that is the crucial parameter. As an estimate of the laser frequency change per data point, the derivative of the relative laser calibration function with respect to the sampling point index is used. The initial scan is taken as a reference and subtracted from all the subsequent calibrations to estimate the stability of the frequency spread per point. These data are shown in Fig. 4b. The relative wavenumber calibration appears least reliable near both ends of a scan, where the polynomial fitting is less

constrained, in particular where low LO power does not produce clear extrema. Otherwise, when these edge effects are removed, the one sigma statistical spread of the data point spacing is ~0.7 MHz/point over the 14-hour period, which is an outstanding stability. To maintain confidence in the frequency calibration, setting a stability requirement of <3 MHz/point, a daily batch of spectra are automatically trimmed following this methodology to match the 3 MHz/point stability requirement on average.

A perhaps more intuitive method to assess the uncertainty in frequency calibration consists in evaluating the relative spectral line positions of the $H_2O$ and the $CO_2$ transitions after calibration (Fig. 4b, inset). To this aim, the peak centroid is determined by fitting Lorentzian lineshapes to the $H_2O$ absorbance spectra, and retaining only those centroids where the summed squared residuals of the fit remain below a threshold. For the day used in Fig. 4, this criterion is met for 333 out of 455 spectra; values differ mostly for low Sun elevation angles when lines are saturated and centroids are least reliable. Using

the $CO_2$ transition as a reference, the standard deviation of the $H_2O$ line position thus estimated is 0.0015 cm$^{-1}$. The median lies around 953.36359 cm$^{-1}$, and if the line position in HITRAN (953.36743 cm$^{-1}$) can be taken as a more reliable reference, this yields a bias of -0.0038 cm$^{-1}$. Likewise, for the 3029 spectra in the archive passing quality control, spanning 8 months of measurements, 2872 successfully had line centroids fitted, resulting in only a slightly higher standard deviation of 0.0024





cm$^{-1}$ and bias of -0.0043 cm$^{-1}$. Overall, the frequency calibration essentially remains very stable and no systematic drift is observed.

### 4.3 LHR instrument performance

In order to characterize the radiometric performance of the LHR, measurements using a calibrated blackbody source are

needed. To that end, a miniature cavity blackbody source with an emissivity better than 0.99 was used to fill the LHR input aperture. In this case, the radiometric power $P$ received by the instrument can be calculated and is expressed by Eq. (8) (Weidmann et al., 2007a), where $R(\nu, T_{BB})$ is the Planck function (spectral radiance) for a blackbody at temperature $T_{BB}$, and $B$ represents the double side band spectro-radiometer resolution. Equation (8) accounts for the polarization sensitivity of the LHR, and for the fact that only a single spatial mode of the received field contributes to the heterodyne signal. It additionally

accounts for broadband absorption between the source and the receiver photomixer through a transmission term $\kappa$:

$$P = \frac{1}{2} \cdot \kappa \cdot R(\nu, T_{BB}) \cdot \frac{c^2}{\nu^2} \cdot B, \tag{8}$$

Under the ideal limit of a shot noise-limited LHR, the noise equivalent power NEP is expressed by Eq. (9), in which $\eta$ stands for the heterodyne efficiency of the photomixer, $\tau$ the integration time, and $h$ denotes Planck's constant. From Eq. (8) and (9), the theoretical shot noise-limited SNR of the LHR can be expressed and is given by Eq. (10), in which $k$ is the

Boltzmann constant. These equations of NEP and SNR are used to assess the experimental LHR performance when referenced to the ideal shot noise limit.

$$NEP = \frac{h \cdot \nu}{\eta} \sqrt{\frac{B}{\tau}}, \tag{9}$$

$$SNR = \frac{\eta \cdot \kappa \cdot \sqrt{B \cdot \tau}}{exp\left[\frac{h \cdot \nu}{k \cdot T_{BB}}\right] - 1}, \tag{10}$$

The blackbody temperature is set to $T_{BB} = 1323 \pm 1.7$ K. Throughout the optical system up to the photomixer the

transmission of the blackbody radiation ($\kappa$) is 57.4% (53.5% when coupled to the solar tracker). The heterodyne efficiency of the photomixer is taken from the quantum efficiency provided by the manufacturer at this wavelength (0.26). It is worth noting that the photomixer is based on a resonant optical cavity design and is not optimized for the ~950 cm$^{-1}$ frequency. When operating in optimized conditions, heterodyne efficiencies of up to 0.5 have been obtained. The LHR double side band spectral resolution is set to 60 MHz, and the integration time to 50 ms. Under these conditions, at fixed LO frequency (2.86

$10^{13}$ Hz or 955 cm$^{-1}$), the measured SNR is 75, which is only 1.9 smaller than the ideal SNR calculated from Eq. (10). A long temporal record of heterodyne signal was also used for an Allan variance calculation to establish the stability time of the system, which was found to be ~100 s (limited by liquid nitrogen-cooled dewar temperature drifts). This quantity underpinned the choice of the integration time.

Next, a similar experiment was run under more realistic conditions of LO frequency scanning. The LO was tuned over ~1

cm$^{-1}$ by a linear current ramp. For this experiment, the LHR double side band spectral resolution was set to 600 MHz. During


the tuning process, the LO also exhibits power modulation (Fig. 5a). The QCL operating set point was chosen such that the laser is nearly at lasing threshold at the start of the ramp. The evolution of the noise over the course of the tuning can be seen in Fig. 5b. As the laser frequency is scanned, the heterodyne noise increases by ~60% (note that the laser frequency is scanned towards lower frequency as current increases). However, since the increase in LO power boosts the heterodyne

signal significantly, the SNR increases and sets at half of the ideal shot noise limited SNR ($SNR_{SN}$).

## 5 Atmospheric measurements

### 5.1 Atmospheric spectra

The LHR has been recording atmospheric transmission spectra since 21 May 2015 and continues to operate whenever the weather allows. Some of the atmospheric transmission spectra recorded with the highest SNR in this period are shown in

Fig. 6. Unless otherwise stated, total acquisition time is 90 s. In the previous section, it has been established that the instrument operates within roughly 2 times the ideal shot noise limit. This represents the intrinsic instrument performance independent of any received radiometric signals. However, the absolute SNR also scales with the amount of power received from the source. Therefore, the highest SNR is obtained on days where the broadband atmospheric extinction (primarily due to aerosols) is lowest and the total atmospheric column traversed (set by date and time of day) is shortest. The spectra of Fig.

6 correspond to this situation. The spectral resolution is primarily set by the double side bandwidth $B$ of the RF filter used (60 MHz or 0.002 cm$^{-1}$ and 600 MHz or 0.02 cm$^{-1}$, respectively). The RF filters govern the instrument line shape (ILS), which has been accurately measured electronically, and which is convolved with the modelled spectrum. The smoothing effect of a larger ILS on the narrow tip of the $CO_2$ absorption line is illustrated in the atmospheric spectra of Fig. 6. Lock-in integration time and sampling frequency have been set such that the spectral resolution determined by the RF filters is

slightly oversampled.

Between 4 June 2015 and 20 January 2016, 6901 spectra have been collected. Semi-automated unsupervised pre-processing (in IDL) is used to load data and metadata, perform the spectral frequency calibration, trim the spectra, remove the broadband power offset, estimate measurement noise and SNR, interpolate the spectra onto a regular grid, compute mean Sun elevation angle as well as for quality control (QC) screening. An initial guess of the polynomial baseline corresponding

to the LO power modulation is also done as an input to the retrieval algorithm. QC is currently based around a set of a priori criteria and thresholds applied to statistical and signal analysis indicators. It is designed to flag up spectra adversely affected by cloud contamination, discontinuous (off-centred) solar tracker pointing and external noise interference. Screening classification errors have been assessed to remain below roughly 10%. From the 6901 original spectra, 3029 have been retained after QC.

Following the method outlined in (Weidmann et al., 2007a), measurement noise $\sigma_{noise}$ (expressed as a heterodyne signal voltage) and SNR are estimated from the one sigma standard deviation and average signal intensity on discrete subsections of the spectra (upper, mid and lower, Fig. 6). For each subsection, the piecewise signal is first corrected for baseline



variation by a $2^{nd}$ order polynomial fit to the data. As mentioned above, the SNR typically decreases from the upper (largest signal) towards the lower part of the spectrum. Within the retrieval algorithm, an average of the noise across the 3 values is currently introduced. Hereafter, the focus is on the most representative mid-section. Since measurements have been recorded with different filter bandwidths (spectral resolutions) and integration times, SNR and noise are normalized by $\sqrt{B \cdot \tau}$ and

$\sqrt{B/\tau}$, respectively, following Eq. (10) and (9), to characterize the instrument's spectro-radiometric performance independently of instrument settings. These normalized values, for the entire archive of 3029 valid measurements, are plotted in Fig. 7a and b, colour-coded according to the acquisition day.

Besides a set of outliers corresponding to the most recent measurements in January, all noise measurements collapse neatly around a median value of $\sim 3 \cdot 10^{-7}$ mV$_{het}$ Hz$^{-1}$, independently of date and time (Fig. 7b). This noise level inherently

characterizes the LHR as an instrument, independently of any received radiometric signal. Conversely, since the SNR varies linearly with signal magnitude, and since the latter is highly dependent on the atmospheric state and the Sun's position, the SNR (Fig. 7a) predominantly reflects environmental conditions. The SNR strongly varies on a seasonal basis, changes with the atmosphere's broadband extinction and cloud cover, and on an otherwise clear-sky day (30 June 2015, red circles), follows a distinctive diurnal pattern. Some sequences exhibit extreme measurement-to-measurement variability; this is most

likely due to fractional cloud cover with a large amount of high-altitude cirrus (Ci) and haze which reduces the signal without blocking it owing to the reduce scattering in the TIR. On 11 September 2015 (orange stars), for example, SNR and solar broadband are strongly correlated ($\rho=0.88$), and the latter, characterizing cloud cover, heavily fluctuates in time.

By converting the noise-level heterodyne voltage into radiometric power units, the normalized noise equivalent power (NEP) can directly be compared to a normalized version of the theoretical shot noise limit from Eq. (9), i.e. $h\nu/\eta$. Therefore, an

instrument calibration curve, mapping heterodyne voltage into radiometric power (radiant flux) units, was established. Towards its low-power end, this curve was constructed by recording heterodyne voltages for cavity blackbody measurements with temperatures varying between 373 and 1323 K, and using Eq. (8) to model the associated radiant flux, assuming an emissivity of 1. Thereafter, for the high-power end, the solar signal was coupled back into the instrument, introducing an unknown radiant flux. Using 23 μm Mylar® polyester sheets as known attenuators (previously characterized by FTIR

measurements), the measured heterodyne signal received was lowered until it reaches the domain covered by the laboratory blackbody calibration, hence calibrating the unknown radiance received by the instrument. Attenuating sheets were subsequently removed one by one to produce calibration points. The radiometric calibration curve was found to be linear. The intercept was forced to zero and the slope was measured to be $G_{v2r} = 0.989 \mp 0.008 \text{pW/mV}$, considering the blackbody measurements only.

Using the radiometric calibration gain, the normalized NEP can be calculated from the data shown in Fig. 7b. These are plotted in Fig. 7c, but this time as a function of measurement date and time. The median noise level exceeds the shot noise limit by roughly a factor of 4. Measurement noise did not significantly drift in time over the 8 months of operation, besides a unexplained sharp increase in January 2016, currently attributed to observed excess low frequency electrical signal picked up



by the detection chain. Compared to the previously established factor of 2 for blackbody measurements, the most likely candidate for this additional performance degradation is broadband thermal noise that is not rejected by the RF filter. This hypothesis is strengthened by the observation that noise seems to increase with the radiant flux received. Future use of a narrower band-pass filter may remediate this issue. Nevertheless, the system consistently operates close to the shot noise

limit, highlighting the excellent radiometric performance of the LHR.

High and medium altitude optically thin clouds (cirrus and some altocumulus) were often found to be transparent enough in the thermal infrared to allow the LHR to collect spectra. The narrow FOV and the high temporal resolution of the LHR, combined with the passive tracking approach, allows the capture of data not only through the clear sky between patchy clouds, but also through thin clouds, as can be seen in Fig. 8a, in which a picture of the sky at the time of measurement is

also shown. Obviously, the signal attenuation due to clouds lowers the SNR, but information can nevertheless be obtained. During the acquisition time of the spectra, (90 s in this particular case), one needs to ensure that cloud attenuation variation will not create spectral artefacts. The broadband NIR solar intensity signal can be used to determine whether this condition is fulfilled, although the non-linear influence of clouds on radiative transfer (Fig. 8a, inset) warrants the exercise of caution. For a specific example, Fig. 8a, inset, shows a fixed frequency temporal record of both the heterodyne signal and the

broadband NIR solar intensity signal as clouds were passing through the LHR's line of sight. Within the linearity region (broadband NIR signal > 0.1V), while the NIR infrared broadband signal is reduced 6 fold, the heterodyne signal decreases by only a factor of ~1.3.

A field- or space-deployable TIR LHR should eventually utilize a thermo-electrically-cooled (TEC) as opposed to a cryogenically-cooled photomixer. To close this technical study, an atmospheric spectrum with a TEC detector (Vigo PV-

4TE-10 with a custom preamplifier) is shown in Fig. 8b, in comparison to an equivalent spectrum with the liquid nitrogen-cooled photomixer. The narrower bandwidth of the TEC-detector, apparent from the sharper $CO_2$ absorption peak, needs accounting for when comparing the performance of both in terms of NEP. After factoring in this difference, it appears that the TEC-detector only underperforms the cryogenically-cooled photomixer by a factor of roughly 2. It is anticipated that more advanced detector arrangements will improve this performance even further.

**5.2 Towards retrievals**

Ultimately, low measurement noise should translate into the small uncertainty required for $CO_2$ flux inversions and carbon cycle studies, provided all biases can be accounted and corrected for. Bias analysis, a detailed description of the retrieval scheme, its configuration, optimization and validation, are beyond the scope of this study and are subject to follow-on work. However, in order to characterize the instrument error propagation and the associated measurement precision and to

demonstrate the approach, preliminary retrievals have been run on some of the screened and pre-processed data collected on 30 June 2015.

The retrieval algorithm remains the same as that used as OSS, with synthetic spectra replaced by real measurements. Input temperature and pressure profiles from the MIPAS mid-latitude reference atmosphere have been substituted up to a height of



~15 km by operational radiosonde data, acquired at 9:00 UTC on the measurement day from station 3743 (51.20°N, 1.80°W). The $CO_2$ a priori profile was in a first instance simply set to a constant 400 ppm VMR up to 30 km altitude, with arbitrarily chosen uncertainties of 50, 30, 10 and 50 ppm at 0.4, 2, 10 and 30 km retrieval grid altitude, respectively, reflecting large variability in the PBL and an unknown concentration in the stratosphere. In reality, $CO_2$ climatologies,

stratospheric age of air empirical models from in situ measurements, and chemistry transport models imply that stratospheric $CO_2$ decreases with height above the tropopause (Andrews et al., 2001; Chatterjee et al., 2013; Saito et al., 2011; Wunch et al., 2011); in future algorithm iterations, the $x_a$ profile and associated uncertainties should reflect a better-constrained state and its variability. The $\mathbf{S_a}$ block matrix corresponding to $CO_2$ was further constructed using an approximate and coarse interpolation of the $CO_2$ correlation matrix given in (O'Dell et al., 2012) (their Fig. 2) onto the retrieval grid, in order to

constrain the retrieved profile to a smoother vertical shape. A relative error of 50% was assigned to all the levels (0.3, 1.5, 3 and 8 km) of the prior $H_2O$ profile and of 100% to the 3 coefficients describing the second order polynomial baseline (RADSCALE) accounting for LO power modulation and broadband atmospheric attenuation. Off-diagonal terms on $\mathbf{S_a}$ corresponding to $H_2O$ and the baseline were all set to 0, as were those on the instrument noise covariance matrix $\mathbf{S_\varepsilon}$. Note that a priori profiles and uncertainties do not currently represent suitable inputs to produce reliable geophysical output, and

will eventually need to be determined from appropriate climatologies and re-analysis data, as emphasized above. The algorithm solution is defined to have converged when $\Delta\chi^2/\chi^2 < 0.001$, within an upper limit of 10 iterations.

Preliminary retrieval results are shown in Fig. 9; only a subset of inversions complying with an arbitrary threshold of $\chi^2/m$ of 2 (Eq. (1)), where $m$ is the number of sampling points (degrees of freedom of the signal), have been retained. The total column-averaged mole fraction of $CO_2$ (in dry air), $TC_{CO2}$ ($X_{CO2}$), remains relatively stable over the course of a day (Fig. 9b),

in particular when a nine point smoothing average is taken over individual data points obtained from a 90 s measurement. The daytime average $X_{Co2}$ measured was 395.85 ppm, which is roughly consistent with GOSAT (interpolated) FTS SWIR L3 data over the UK (~400 and ~396 ±2 ppm (standard error) on average for months of June and July 2015, respectively).

The uncertainty on $X_{CO2}$ from error propagation (1.9 ppm for a 90 s measurement at 60 MHz resolution) comprises a contribution from the (arbitrarily chosen) uncertainty on surface pressure of 3 hPa. If the latter is ignored to capture only

instrumental error propagation, the uncertainty on $X_{CO2}$ mirrors that of $TC_{CO2}$, estimated to be ~1.5 ppm. As already mentioned, no systematic errors or bias analysis is included at this stage. This uncertainty is just above two times worse compared to that obtained using the OSS at 60 MHz resolution (0.7 ppm), and within the requirements of $CO_2$ flux inversion modelling studies. The comparison has to be treated with caution as the OSS used to identify optimum sounding narrow windows was constrained with a larger a priori uncertainty (20% relative error on all levels) and without covariance terms on

$\mathbf{S_a}$. We anticipate these uncertainty figures to be improved by increasing the SNR of the recorded spectra (higher spectral resolution and/or longer integration time than 90 s). Introducing more rigorous a priori profiles from climatologies, with adequately characterized covariance matrices is also in scope for follow on improvements. Here, the large covariance terms introduced in $\mathbf{S_a}$ for $CO_2$, meant to smoothen the profile by emulating large-scale free tropospheric vertical mixing, obviously shifts the dominant error contribution from measurement error in the OSS (Fig. 2a) to smoothing error (Fig. 9a).



The water vapour total column is better suited for a straight side-by-side comparison between the OSS and actual measurement instrumental error propagation. In both cases, a priori constraints were identical. The OSS indicated a 1% relative error on $TC_{H2O}$, whilst the measurements are at a 5% relative error level, compared to the 50% relative error assigned to the prior humidity profile on all state vector altitudes. Although a difference by a factor of 5 appears consistent with the

average noise degradation of real measurements compared to the shot noise limit (~4), this comparison is not straightforward, given different humidity profiles and different (and dominant) smoothing error contributions, particularly in the boundary layer.

Unlike total columns, vertical profiling is, unsurprisingly given the a priori, under-constrained and shows unphysical oscillations. Follow-on work will focus on addressing these issues to better condition the inversion process using valid a

priori and improved geophysical inputs.

## 6 Conclusions and future work

This paper reports the technical development, demonstration and performance assessment of a benchtop prototype thermal infrared (953.1 cm$^{-1}$) quantum cascade laser heterodyne radiometer (LHR) coupled to a passive Sun tracker designed specifically for total column and vertical profile measurements of $CO_2$ in solar occultation configuration. An observing

system simulator was used to estimate the potential performance and define an optimum narrow spectral window to be used by the instrument. Data from over 8 months of operation in 2015 near Chilton, UK, confirm that atmospheric measurements with noise levels down to ~4 times the shot noise limit can consistently be achieved. This provides the opportunity for taking very high spectral resolution (here, 60 MHz/0.002 cm$^{-1}$ and 600 MHz/0.02 cm$^{-1}$, corresponding to resolving powers of ~500,000 and ~50,000) measurements with high signal-to-noise ratios (median of 113 at 60 MHz, median of 257 at 600

Mhz) at a relatively high temporal resolution (here, 90 s). Spectral information is recorded over a narrow spectral window of ~1 cm$^{-1}$, focusing on the joint measurement of water and $CO_2$ ro-vibrational transition lines to enable the conversion of the total column-averaged $CO_2$ mixing ratio into a dry air equivalent. A narrow window, conversely, limits the risks of cross-talk by other trace gases, and single transitions make spectroscopic error relatively easy to deal with. Preliminary retrieval results aiming to investigate the instrumental error propagation into total column-averaged mixing ratios show an $X_{CO2}$ error in the

range of 1-1.5 ppm. Biases and error propagation from geophysical inputs have not been considered yet.

The demonstrated instrumental error reveals that the LHR promises to provide good quality $X_{CO2}$ data in relation to satellite data validation and carbon cycle studies, especially as the current instrument's performance can be improved through longer integration times and improved detection schemes.

The inherently narrow field-of-view (FOV) of a LHR, combined with short acquisition times, was found to increase the

probability of collecting unobstructed spectra on cloudy days. Owing to operation in the TIR, signals with clear spectral signatures can still be acquired through optically thin clouds (cirrus) and haze, although at the cost of a reduced SNR in line with the additional attenuation.



From this first demonstration and instrumental assessment, several pieces of follow on work are required. On the data retrieval side, thorough algorithm characterization and sensitivity (e.g. through input perturbation) studies are needed, alongside more rigorously-chosen auxiliary and a priori data to improve the retrieval of, and confidence in, the $CO_2$ profiles and total column amounts and uncertainties. This includes comprehensive error and bias evaluation, validation through

cross-comparison with collocated and simultaneous measurements with other ground-based remote sensing and in situ instruments as well as satellite and/or airborne observations, and retrievals with other independent algorithms. In a next iteration of the retrieval algorithm, atmospheric broadband attenuation will also tentatively be extracted as a by-product. The development of an operational retrieval scheme will require a number of incremental improvements to refine the algorithm, and could follow steps similar to those taken for the TCCON network.

On the instrument development side, the full TIR LHR potential is to be realized through size reduction and engineering into a compact, rugged and autonomous package for increased portability to ease field deployment. Miniaturization has already started using hollow waveguide hybrid optical integration (Weidmann et al., 2011b) to open up the path towards incorporation of the TIR LHR into dense and large-scale autonomous instrument networks and deployment on air- or spaceborne platforms.

**7 Copyright statement**

The works published in this journal are distributed under the Creative Commons Attribution 3.0 License. This licence does not affect the Crown copyright work, which is re-usable under the Open Government Licence (OGL). The Creative Commons Attribution 3.0 License and the OGL are interoperable and do not conflict with, reduce or limit each other.

**8 Code and data availability**

Code and data are not yet formatted into an operational package. Currently available versions are available upon direct request to the authors. In the future, once properly formatted with inclusion of quality assurance parameters and relevant metadata, they may be made available through for example the British Atmospheric Data Centre.

**9 Author contributions**

AH prepared the manuscript, operated, and contributed to the development of the instrument and its software, wrote and
upgraded new data pre-processing and existing retrieval source code, and performed the simulations and data analysis. NM performed the characterization and testing of the instrument, and contributed to the manuscript, instrument operation, data analysis, and with expert advice throughout the project. MH designed and assembled the first optical layout of the breadboard and of a new laser cooling system, and performed the characterization of the local oscillator. DW conceived the





instrument and study, designed the solar tracker, and contributed to the data analysis and manuscript, and provided technical lead throughout the project. All authors revised and approved the manuscript.

## 10 Acknowledgments

This work was supported through a Centre for EO Instrumentation and Space Technology (CEOI-ST) grant (contract n°
RP10G0327C20). The authors wish to thank Gary Williams at RAL Space for his technical assistance and contributions to the design, development and implementation of the solar tracking system, Jolyon Reburn for revising the manuscript and the members from the Remote Sensing Group for help and useful discussions. A part of the solar tracker control software was built around an open-source Sun position LabVIEW application written at the CRP Henri Tudor, Luxembourg; radiosonde data was downloaded through the University of Wyoming web portal.

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





**Table 1: Summary of column measurements and errors from the OSS. Values under the error propagation header correspond to the first ensemble member. The planetary boundary layer (PBL) column is here defined as between 0-1.2 km.**

| Quantity | Truth | 60 MHz spectral resolution | | 600 MHz spectral resolution | |
|---|---|---|---|---|---|
| | | Error prop. | Statistical ($\mu \pm \sigma$) | Error prop. | Statistical ($\mu \pm \sigma$) |
| Info content $H$ [bit] | | 50.2 | | 61.7 | |
| DFS $d_s$ [-] | | 7.9 | | 8.2 | |
| $CO_2$ column [ppm] | 380.52 | $379.16 \pm 0.64$ | $380.19 \pm 0.81$ | $380.79 \pm 0.44$ | $380.31 \pm 0.50$ |
| $H_2O$ column [ppm] | 1920 | $2093 \pm 19$ | $2099 \pm 9$ | $2057 \pm 35$ | $2056.3 \pm 1.5$ |
| $X_{CO2}$ [ppm] | 380.69 | $379.36 \pm 1.29$ | | $380.98 \pm 1.21$ | |
| $CO_2$ PBL column [ppm] | 367.91 | $356.94 \pm 16.48$ | $357.26 \pm 17.78$ | $366.58 \pm 7.53$ | $366.13 \pm 7.95$ |



**Table 2: Laser tuning coefficients corresponding to the model in Eq. (7).**

| | |
|---|---|
| Wavenumber offset $[\text{cm}^{-1}]$ | $\sigma_0 = 955.90 \pm 0.05$ |
| Linear current tuning coefficient $[\text{cm}^{-1}\text{A}^{-1}]$ | $i_1 = -1.71 \pm 0.2$ |
| Quadratic current tuning coefficient $[\text{cm}^{-1}\text{A}^{-2}]$ | $i_2 = -5.42 \pm 0.2$ |
| Linear temperature tuning coefficient $[\text{cm}^{-1}\text{K}^{-1}]$ | $t_1 = -0.0539 \pm 0.0011$ |
| Quadratic temperature tuning coefficient $[\text{cm}^{-1}\text{K}^{-2}]$ | $t_2 = (-5.56 \pm 0.29) \times 10^{-4}$ |
| Current temperature tuning coupling $[\text{cm}^{-1}\text{A}^{-1}\text{K}^{-1}]$ | $x = -0.020 \pm 0.002$ |



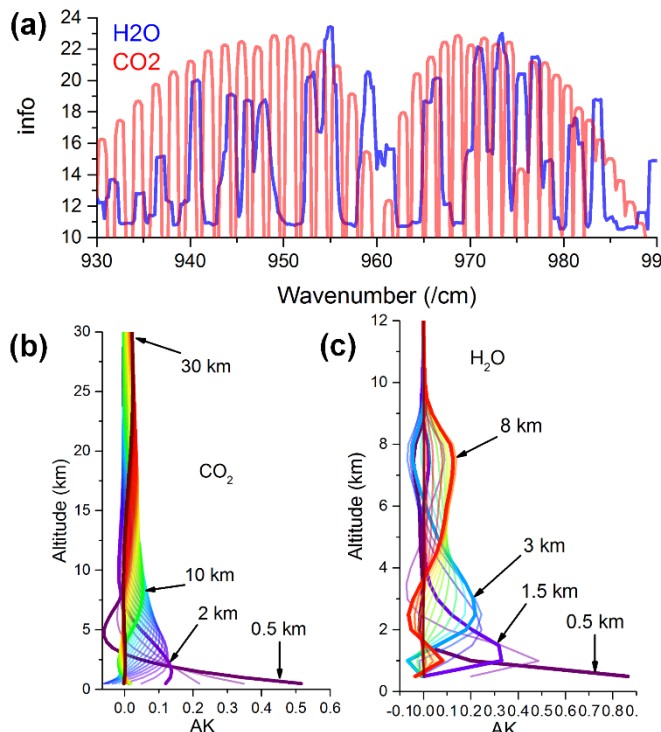

**Figure 1: (a)** Observing System Simulator-retrieved Shannon information content for $CO_2$ and $H_2O$ for a narrow spectral window of 1 cm$^{-1}$, successively slid by 0.2 cm$^{-1}$ over the full spectral range; **(b)** $CO_2$ and **(c)** $H_2O$ high-resolution averaging kernels computed for the finally selected narrow spectral window. The 4 thick lines pointed at by arrows correspond to the state vector heights used in the instrument performance simulation (Fig. 2).





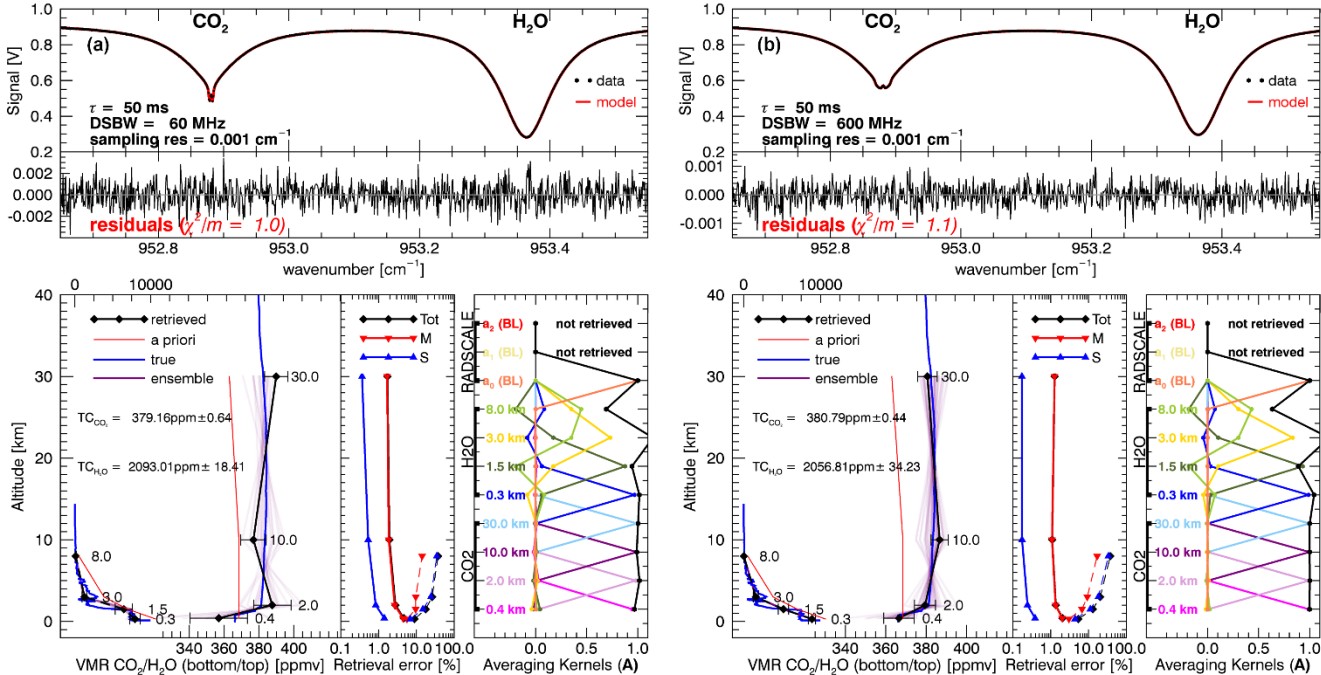

**Figure 2:** OSS simulation for the selected spectral narrow window under the assumption of a shot noise-limited operation of the LHR with a 60° Sun elevation. Instrument spectral resolutions are set to (a) 60 MHz and (b) 600 MHz. The top panel shows a synthetic spectrum (data) with noise, calculated from a known "true" state of the atmosphere. Superimposed is the retrieved
5   spectrum (model), and the difference is shown as residuals below. The bottom panel depicts all (purple) and one selected (black) amongst the retrieved state ensemble members as separate $H_2O$ and $CO_2$ profiles, in comparison to the true and a priori state (left). Error bars characterize the retrieval uncertainty at the state vector altitudes. The estimated total columns (TC) with uncertainties are given as text. Total retrieval errors (Tot) and contributions from measurement noise (M) and smoothing error (S) are shown as profiles in the centre. The averaging kernels for the composite state vector are shown on the right; RADSCALE
10  denotes the instrument baseline (BL), which corresponds to a 1:1 mapping from radiometric into signal voltage for these simulations.



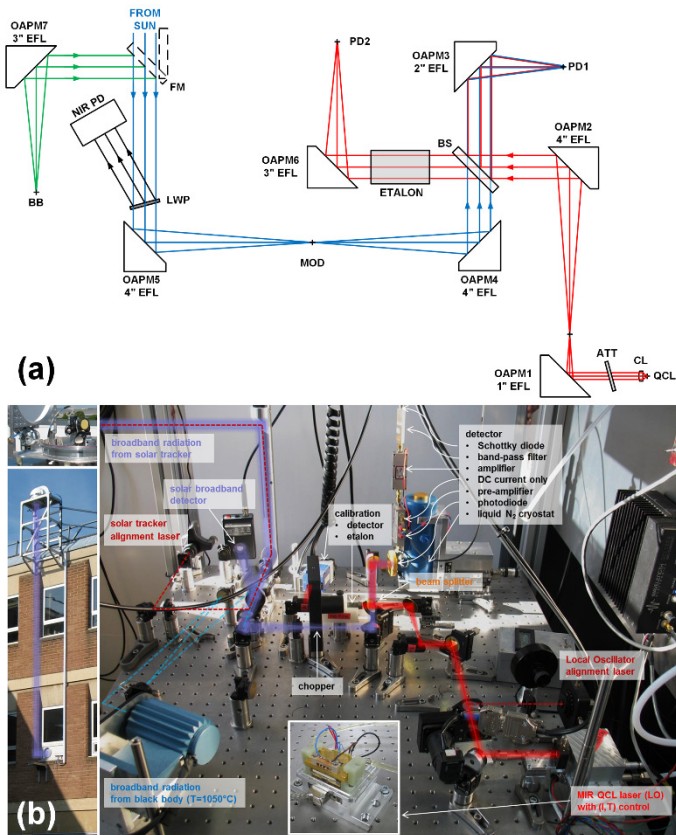

**Figure 3: (a) Schematic of the LHR optical layout with the optical path of the LO in red and Sun broadband radiation in blue; (b) annotated picture of the breadboard prototype, next to the outdoor wall-mounted structure with optical path used for solar tracking.**





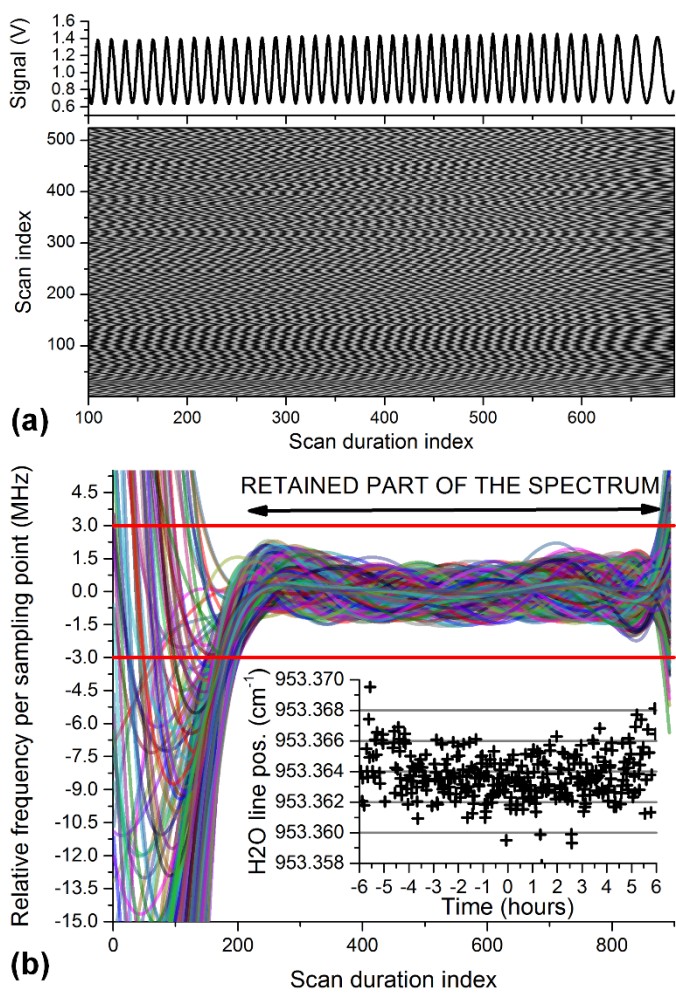

5   **Figure 4: (a)** Contour plot showing a subset of successive etalon signals (corrected from envelope power variation, lower panel). The upper panel shows one signal trace; **(b)** consecutive derivatives of the same day's measurements' relative frequency vectors, from which the initial derivative vector has been subtracted to estimate frequency calibration stability. The inset shows the free-floating $H_2O$ spectral line centroid position for the same batch of measurements.





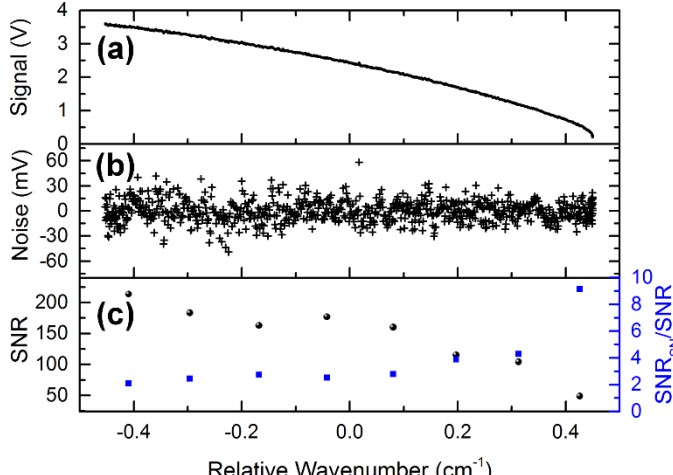

**Figure 5: (a) Heterodyne signal modulation throughout the LO frequency scan using a 1373 K blackbody as a source; (b) heterodyne signal noise over the laser tuning, and (c) evolution of the LHR SNR computed on successive chunks of 100 data points. $SNR_{SN}/SNR$ depicts the ratio of the theoretically-calculated shot noise-limited SNR (Eq. 10) over the measured SNR.**





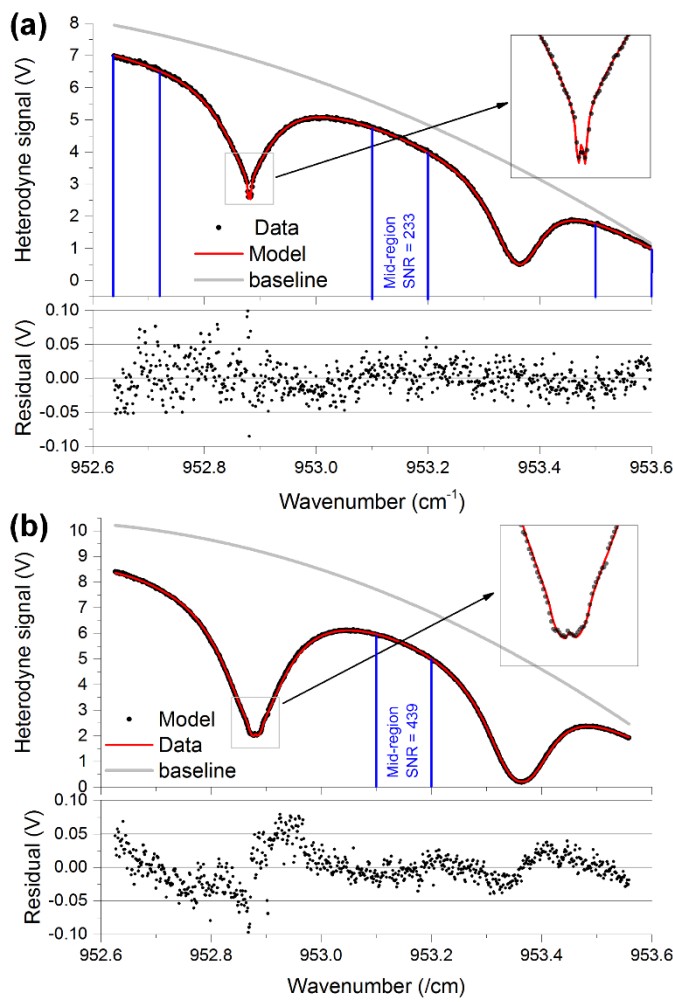

**Figure 6: Atmospheric spectra (and modelled fits after retrieval) in the selected window at two spectral resolutions, recorded (a) on 10 September 2015 11:17 UTC at 60 MHz with a mid-section SNR of 233, and (b) on 28 September 2015 15:29 UTC at 600 MHz with a mid-section SNR of 439. The thin blue lines delimit the spectral window over which the SNR and measurement noise are estimated after applying a second order polynomial fit. The thin grey lines show the polynomial baselines due to LO power modulation.**





**Figure 7:** LHR radiometric performance analysis over the full archive of quality-controlled spectra: (a) normalized mid-section (953.15 ± 0.05 cm$^{-1}$) SNR; (b) normalized mid-section one standard deviation noise in heterodyne voltage units with histogram distribution; (c) normalized noise converted into a radiant flux as a function of date and time of measurement. The red solid and broken lines correspond to the theoretical shot noise limit and consecutive multiples thereof.



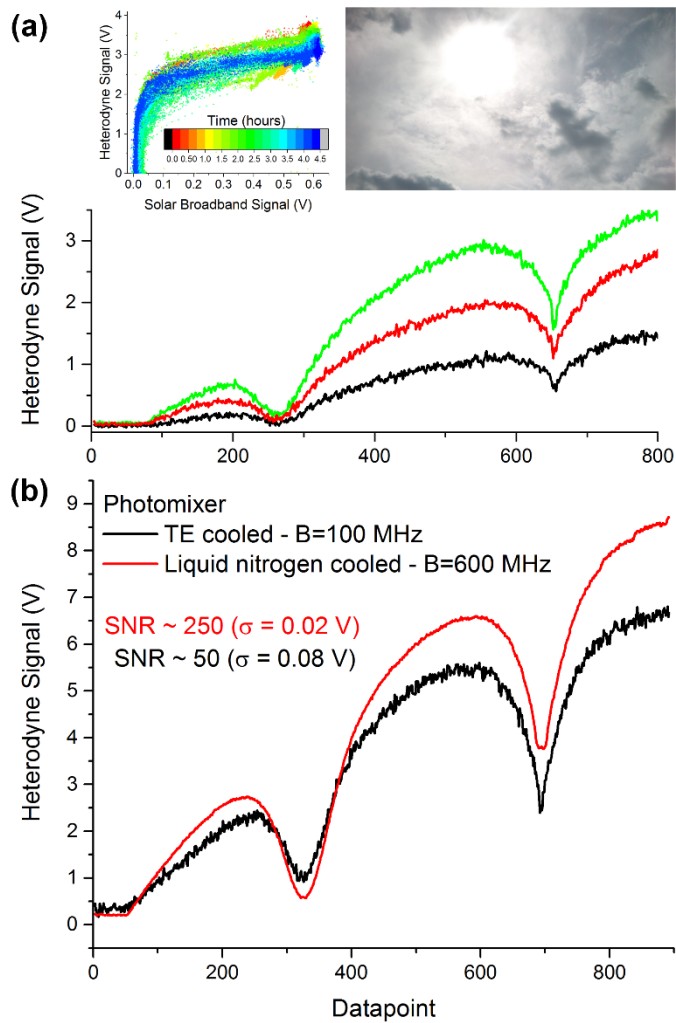

**Figure 8: (a) LHR atmospheric spectra recorded for three different levels of cloud attenuation (at 60 MHz spectral resolution and with 50 ms integration time). The insets show a picture of the sky during the measurements, and a scatterplot of NIR solar broadband versus heterodyne signal, constructed using a number of constant frequency recordings with variable cloud amounts and types; (b) spectra obtained with two different types of photomixers. The liquid N$_2$-cooled photomixer with 600 MHz double side bandwidth has primarily been used for this work, and is compared to a thermo-electrically (Peltier) cooled photodiode with 100 MHz double band bandwidth.**





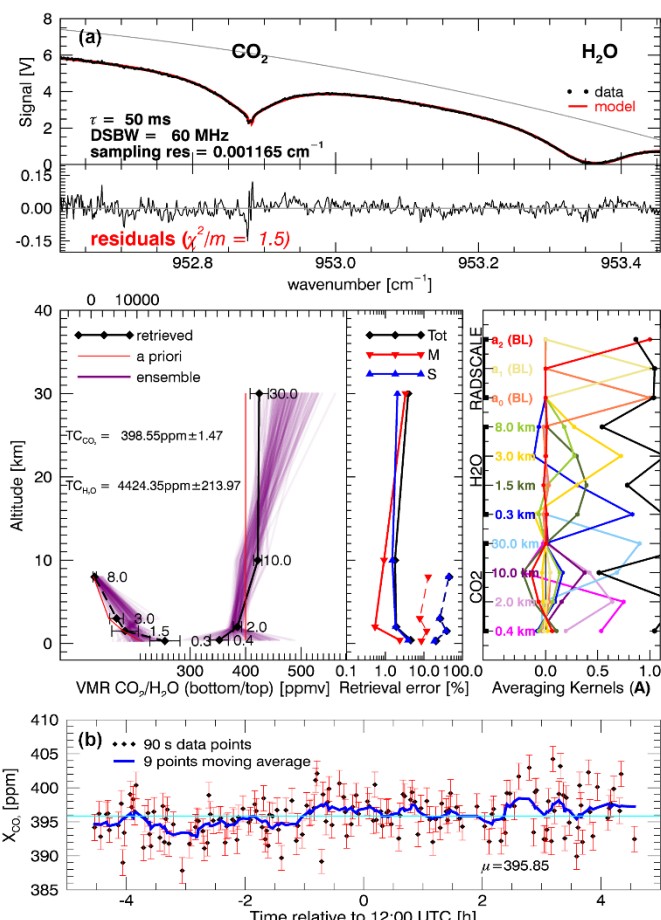

**Figure 9:** Retrieval results for measurements acquired on 30 June 2016; only the 166 out of 455 retrievals that converged within 10 iterations and where $\chi^2/m < 2$ have been retained: (a) same as Fig. 2 but for real data and with different prior conditions (see text). Purple profiles correspond to the ensemble of retained inversions; the selected profile corresponds to a measurement at 12:05 UTC; (b) associated daily evolution of $X_{CO_2}$ and moving average computed over the individual retrievals.