# Peer review of "Thermal infrared laser heterodyne spectro-radiometry for solar occultation atmospheric CO2 measurements"

_Atmospheric Measurement Techniques, 2016_

## Short Comment (SC1) · 14 Jul 2016

Dear authors,

thank you for your manuscript. As a TCCON PI (Ascension Island), I have some specific comments about statements and claims that you make with respect to TCCON:

**General comment on XCO2**

For TCCON, the observation of O2 lines is essential as a proxy for airmass. Since you cannot observe O2, how do you determine airmass? Deriving airmass from surface pressure alone is considerably less accurate and has therefore been depreciated in TCCON for years.

**p. 3, l. 18-19: "Whilst TCCON currently provides the best benchmark in ground-based remote sensing of CO2, the upfront investment in establishing a site and the subsequent running costs prevent the network from reaching a high density."**

I often hear this argument - mostly from people outside the TCCON community. However, I think it is flawed because I don't see how a (potential) network of different instruments could do fundamentally different. First of all, the main limitation in the expansion of TCCON is not the cost of the FTS itself but rather the operational cost of running a site in a remote place. The not-so remote places like Europe and North America are already well covered. The typical carbon-cycle-related footprint of a total-column instrument is so large that a handful of instruments per continent are enough.

However, even in a place with good infrastructure the operational cost (excluding personnel) is in the range of 10% of the FTS acquisition cost per year. In a remote place like Ascension Island this is more like 20–25%. Main cost drivers are utilities, data transmission, spare parts and maintenance visits. If you include personnel costs in the calculation, the initial investment for the FTS would probably only account for 10–15% of the total cost for running a site for 10 years. This will not be fundamentally different for other types of instrumentation. Many TCCON instruments are built into standard 20-foot shipping containers that are easy to move to any place in the world that has road access. Despite extremely harsh environmental conditions, the Ascension Island instrument has been running up to 10 months in a row without a maintenance visit. Just by exchanging the instrument inside a lab container, the basic cost model for running a remote site would not be changed substantially. It would only be different for lightweight instruments that can run autonomously on solar power, have very limited data upload needs, run without consumables and do not require regular maintenance by qualified personnel.

BTW: just the need for cryogenic cooling for your QCL's photodiode (p. 9, l. 15) would already be a show stopper for most remote sites. So replacing it with a TEC would be very beneficial.

**p. 8, l. 32: "The degrees of freedom for signal imply that 8 (or possibly more) independent pieces of information can be retrieved from a spectrum."**

I find it hard to believe this number. Your instrument seems to be comparable to a TCCON instrument in spectral resolution, signal-to-noise ratio, acquisition time. It is certainly not an advantage that your bandwidth is only 0.025% of a TCCON spectrum. So I wonder why similar profile retrievals for TCCON only yield 3 degrees of freedom (Connor et al., doi:10.5194/amtd-8-12263-2015). Could it be that your OSS is assuming very idealized atmospheric and measurement conditions? I assume you are not taking into account real-world effects like horizontal atmospheric gradients, the changing airmass during the 90-sec-measurement, non-perfect instrument lineshape, non-Gaussian noise etc.?

**Section 3.2: "Passive Solar Tracker"**

In TCCON data, we see a lot of effects that are related to tracking accuracy - especially to the exact position on the solar disc. The Fraunhofer lines in the broadband FTS spectra can be used to correct the effects of pointing inaccuracies. However, your instrument cannot see these lines. In TCCON, the gold standard for solar tracking is camera-based tracking of the solar disc image directly on the entry aperture at high frequency. This significantly improves the measurement precision compared to quadrant-diode tracking or passive tracking (not used).

---

## Editor Comment (EC1) · F. Hase (Editor) · 10 Aug 2016

In supplementation of the points raised by D. Feist in his previous comment, I would like to add the following remarks: (1) The comment wrt to airmass reference and pointing accuracy seems important to me. It contradicts common experience that a passive tracker works so precisely as claimed by the authors. The claim should be verified by presenting actual measurements of the solar position on a target during the day. Moreover, the authors should explain which steps needs to be taken (hardware and software) for reproducing this tracking quality if the instrument is duplicated or operated in a campaign mode. (2) A retrieval with 8 degrees of freedom (NDOF) would be an outstanding achievement. As investigated by the FTIR community so far, a profile retrieval for CO2 is demanding (due to low variability of profile shape), even when attempting NDOF of about 2. I understand that the heterodyne approach might outperform the FTIR, but the obstacles met when it comes to profile retrieval are partly of extrinsic origin from the instrumental point of view. Is the spectroscopic description in such an excellent shape that we do not need to worry about spurious profile variability in the retrieval result as function of temperature, water vapor content, airmass, etc? It would be desirable to support the claim by demonstrating e.g. the detection of expected plausible profile variations during a measurement day based on retrievals of real data (increase of boundary layer thickness). (3) I think it is justified to start from an error estimation based on model simulations when pioneering a new observational approach. However, such an estimate should not be overly optimistic. The temperature error should definitely been taken into account, as it cannot be assumed that the temperature profile is known exactly - actually, the temperature uncertainty often is large, especially in the boundary layer variations well beyond 10 K can occur during an observation day (which implies sunny conditions and therefore high insolation). This temperature error might be a critical limitation for the proposed observational approach: it should be recalled that the O2 rationing applied by TCCON does not only compensate airmass errors but also partially compensates the temperature error in the resulting XCO2. (4) The comparison of an average XCO2 value over an extended observation period with a monthly mean of GOSAT observations is not a very useful demonstration. The bias between GOSAT and a mid-IR observation is probably dominated by the systematic error of the assumed linestrength of the CO2 line used for the observation. Instead, it seems crucial to me to demonstrate that the new sensor is capable of detecting the variability of XCO2. If the observational period is too short for detecting the annual cycle and if no data from a collocated reference sensor is available, if finally GOSAT observations are too sparse for a detailed intercomparison, still a comparison with modelled XCO2 data as provided by Copernicus / MACC would be possible.

---

## Author Comment (AC1) · 11 Aug 2016

Dear Dr Feist,

We thank you for your comments and your insightful contribution as a TCCON PI to the discussion on our manuscript related to the LHR instrument development. It is particularly useful to have views from an operational (user) and community perspective.

As a preliminary word of caution, we would like to re-emphasize that it is by no means our intention to suggest that LHR instruments should replace well-established FTIR instrumentation as part of TCCON. Commercial FTIR with its associated experimental protocols, data analysis and traceability developed in the context of TCCON benefit

from a high level of maturity. For this reason, we believe TCCON remains the established benchmark against which it makes sense to evaluate the performance of the LHR during its instrumental development phase. We do emphasize that our retrievals are preliminary and require more analysis and developments (p. 17 l. 27). As with the instrumental development, our data processing maturity is not yet on a par with that of TCCON, which has benefited from a decade of international development (e.g. Yang et al., 2002, doi: 10.1029/2001GL014537). However, we do see interesting potential in thermal infrared LHR systems based on the initial demonstration we presented in the manuscript. This appears to us to be complementary to TCCON, offering potentially light-weight and compact instruments, which may offer interesting trade-offs as far as deployment is concerned (high density, rapid and temporary deployment, remote autonomous operation for instance). It is also worth keeping in mind that the experimental approach to trace gas sensing focused upon in our manuscript is, of course, generic. Hence, LHR can be used for different targeted gases and applications. The stringent requirements within the context of $CO_2$ monitoring provide an excellent opportunity to demonstrate the research instrument capabilities.

Specific replies to your comments follow hereafter.

Comment 1 on XCO2

It is correct that we do not observe O2 and hence air mass. We agree that the implications in terms of accuracy ought to be analysed and quantified. Our initial retrieval efforts rely on auxiliary input data (ultimately from ECMWF analyses, where we intend to use near real time products) for air pressure and temperature profiles, and a surface-level pressure for the dry air column estimation. The current scope of the manuscript focuses on instrumental error propagation. Follow-up studies on detailed error and bias analysis propagating down to XCO2 are planned for the next phase of our endeavour, as emphasized in our conclusions and elsewhere. We do welcome contributions and collaborations for improving our retrieval efforts.

[Figure]

Comment 2 on TCCON network density and running costs

Thank you for sharing your TCCON cost analysis. This is very enlightening, especially since we are in the process of developing one of our Bruker IFS125 FTIR's towards TCCON status at our Harwell site. We propose to change p. 3 l. 19 from '[...] and the subsequent running costs prevent the network from reaching a high density.' to '[...] and the subsequent running costs are high.', which should be factually correct. From our interactions with various funding bodies, and prospective users, we gathered that even an up-front investment of 10-15% of the total cost is still worth taking into consideration. In a very pertinent manner, you mention that the basic cost model for running a remote site would only change substantially for a lightweight, autonomous (and solar-powered) instrument. That is precisely what we aim to work towards with the LHR. The current prototype described in this manuscript is built on an optical table and uses cryogenic cooling for the heterodyne mixer/detector (the QCL is TEC- and water-cooled). We have already demonstrated in the laboratory cryogenic-free operation with a test system about the size of a shoe box. Forthcoming development iterations are planned to be heavily integrated and ruggedized (see p. 20 l. 10). As you mentioned, the use of a TEC-cooled detector is documented in the manuscript (p. 17 l. 18).

Comment 3 on the degrees of freedom (DFS)

We have consistently derived DFS (trace of the AK matrix for all retrieval products combined) values of that order for our simulations, the inputs to which are outlined in the text. A new simulation with the latest iteration of our retrieval algorithm and with high-density optimized grids for $CO_2$ and $H_2O$, produces a DFS of 8.4 (of which ∼1 is due to the a0 baseline coefficient, ∼2.8 relates to $H_2O$ and ∼4.6 to $CO_2$). Yes, the OSS currently assumes 'idealized conditions', in terms of perfectly known temperature, pressure and potential interfering species profiles, a 1D atmosphere, an instrument working at the shot-noise limit, and no forward model error in the retrieval. The instrument noise is assumed Gaussian (p. 7 l. 23), and a rudimentary Shapiro-Wilk normality statistical test is performed during measurement pre-processing to check whether the Gaussian

noise hypothesis must not be rejected. This feature has not yet been fully tested, and has not been mentioned in the manuscript, but preliminary trials seem to imply that Gaussian noise is a mostly valid assumption. The instrument line shape is taken into account and, unlike for other type of spectrometers, can be precisely measured in the electronic domain and is very stable, which is one of the advantages of LHR (p. 7 l. 20, p. 15 l. 16). Indeed, a change of airmass over the 90s measurement is not considered for the purpose of this instrument development manuscript, as this would be part of follow on work related to more advanced data processing and analysis. The manuscript concludes with section 5.2 on a first retrieval of real measurements with real noise (Fig. 9). The DFS for a sample measurement therein is 7.6 (of which 3 are due to the baseline coefficients). For our first retrieval showed in this manuscript, a priori are almost not constrained. The a priori error in Connor et al. (2015) is not specified, it may also account for the difference you raised. The DFS of ~3 by Connor et al. seems to relate to the DFS of CO2 only. Though retrieval performed on CO2 only would need to be performed to estimate an exact figure, the partial DFS for CO2 in Fig. 9 (i.e. the trace of sub-matrix A) amount to ~2.5. Lastly, we would welcome independent cross-checking of our results, e.g. with a different retrieval model, should that be desirable. Sample data or code can be made available upon request.

Comment 4 on the passive solar tracker

We agree and are aware that the tracking system currently implemented for the instrument demonstration is not as accurate as the camera-based tracking recommended for TCCON operation sites. We do mention that '[. . .] better pointing accuracy may eventually be required, especially at low elevation. This, in turn, is likely to be best achievable with a complementary active feedback mechanism.' (p. 12 l. 6.) We have been considering options such as the Sun disc imaging technique described by Gisi et al., 2011 (referenced) for future iterations (the camera-tracking you mentioned). Alternatively, commercial solutions based around this CamTRACKER concept may also be available, and could easily be interfaced to the LHR. The detailed quantification of

pointing accuracies is outside the scope of this work. Again, this relates to forth-coming instrument bias analysis and full error budget. Once pointing errors propagation down to XCO2 have been quantified and compared to other error sources, we aim to ensure the optimum cost/performance trade-off for the pointing system. Since we are not observing solar lines, the measurements are insensitive to the Doppler shift error related to tracker pointing inaccuracies and solar rotation. Ultimately, we believe that it is of the utmost importance to know accurately where we are pointing (providing we remain within the solar disk), rather than pointing to a specific line of sight. The bias and error budget analysis work planned as a next step will provide further quantitative insight.

---

## Referee Comment (RC1) · Anonymous Referee #2 · 17 Aug 2016

This paper is a thesis–like investigation on a technique to measure atmospheric CO2. The described work includes discussions on the motivation, requirements, theoretical modeling, instrumental development with laboratory and field measurements, and the development and use of retrieval and analysis/interpretation techniques. Results, the authors argue, will enable a compact and efficient method for monitoring the important greenhouse gases in the Earth's atmosphere including CO2, water and other atmospheric constituents. The infrared heterodyne technique in the thermal infrared is a powerful technique for extremely high spectral resolution spectroscopic measurements that measure the true shapes of individual spectral lines. Resolved line shapes contain information on the species abundance and temperature in regions probed, as well as

winds which can modify the line positions and shape. The uniquely high spectral resolution enables unambiguous identification and separation of lines of multiple species even in spectrally dense atmospheric wavelength regions. The authors' ultimate goal appears to be to augment and complement ongoing space-based and ground based networks monitoring CO2 and other atmospheric constituents.

Although the authors provide multiple references and discussion on the scientific motivation and current monitoring capabilities and their direct instrumentation heritage, they should address earlier work using infrared heterodyne spectroscopy for Earth and planetary studies. One of the innovations described is the use of quantum cascade semiconductor lasers (QCLs) as local oscillators (LOs). Less capable tunable lead salt semiconductor laser LOs have been used for similar studies in the past (e.g., Frerking and Muehlner, Appl. Optics 16, 526 (1977); Glenar et al. Appl. Optics 21, 253 (1982)). QCL local oscillator use was discussed in Wirtz et al., Spectrochimica Acta Part A 58, 2457 (2002)\ and Sonnabend et al., Journ. of Quantitative Spectroscopy & Radiative Transfer 109, 1016 (2008). Previous measurements and analysis of atmospheric measurements have been discussed in: Menzies and Seals, Science 197, 1275 (1977); Abbas et al. JGR 84, 2681 (1979); Kostiuk, Infrared Physics 35, 243 (1994); and Fast et al. GRL 31, L08109 (2004). Some comparison to these earlier efforts is warranted. Why is their approach better than previous techniques?

On page 2 starting on line 15 the authors describe the importance of retrieving changes in the "lowest 2-3 km" boundary layer, since that is the region of greatest carbon exchange. They seem to imply here and later that the IR heterodyne technique may retrieve a more accurate result. The ultra high spectral resolution is not necessarily a benefit when the line widths are wide as the pressure-broadened line are near the surface. Lines are several GHz wide and the information on the lower atmospheric region is in the wings of the lines and 60 or even 600 MHz resolution are of limited benefit, although the line shapes are not as affected by instrumental functions as in lower resolution spectrometers. Clearly the higher resolution will provide much more

accurate information at higher altitude (lower pressure) regions. A direct comparison of expected ultimate accuracy of their mole fraction/column density retrievals vs. current measurement capabilities would be useful, e.g., comparison to TCCON network retrievals. Some of this may be hidden in the text already.

The modeling of a synthetic spectrum from an optimum instrument and fitting it is a good approach in the paper, testing the retrieval program and ultimate sensitivity of the method. The calculated noise level for the resolution bandwidths of 60 and 600 MHz is correct in the modeling. However, the true spectral resolving element is not the 60MHZ or 600 MHz bandwidth as quoted in the subsequent actual measurements. On page 10, paragraph starting with line 9 it is stated that the RF filters defining the bandwidths span 50-80 MHz and 50-350 MHz on each side of the LO. The heterodyne detection is a double sideband process (the photomixer cannot distinguish frequencies below and above the LO frequency) and in each case the noise and signal from each sideband add to give the sum of two 30MHz (300MHz) signals, hence 60MHz and 600MHz. However, the two sidebands are separated by 100 MHz and sample two noncontiguous frequency regions. Therefore, as the QCL LO is tuned over a spectral line each of the two bands samples a different portion of the line, starting at 100 MHz apart and including signal at frequencies increasing in opposite directions over the prescribed bandwidths, e.g., -50 to -80 MHz and 50 to 80 MHz. This introduces a kind of instrumental function that may affect the measured lineshape and change the effective spectral resolution (not 60 MHz). This effect was not addressed in the paper. A similar question arises in the TEC cooled detector measurements (Fig. 8b). Is the single bandwidth 0-50MHz or 50-100 MHz in both sidebands?

I have difficulty determining what the actual degradation from shot noise limited operation of the LHR is. In measurements with a blackbody a factor of ∼2 is quoted. In field measurement (Fig. 7) a factor of 4 is determined (Abstract and page 16), possibly due to "broadband thermal noise". It would be good to have a list and numbers for the full degradation from ideal performance, including optical losses, detector quantum

efficiency, chopping, polarization...etc. It may be just nomenclature, but shot noise limited operation is affected by losses due to all of these. A total factor of 4 seems too good since several of these alone will contribute a factor of 2.

In summary, aside from the comments above, I believe the authors did a good, but difficult job of developing, modeling, and testing the LHR by executing measurements and analyses of the atmospheric $CO_2$ and $H_2O$. They demonstrated that technological improvements enhanced the thermal infrared heterodyne technique for the study of atmospheric constituents. Their goals of a more autonomous, compact design for LHR capable of measurements from multiple locations monitoring changes in our atmospheric composition could be very useful, particularly as probes of lower pressure regions where the very high spectral resolution has a distinct advantage.

Technical comments/corrections:

Page 8, eq. 6: subscript g's are different. Should this be the case?

On page 10 Lines 4 and 5; it is stated that 0.52 mrad represents the coherent FOV and this is 1/8 of the apparent solar disc angle. The solar disc is $\sim$0.5 degrees, $\sim$8.7 mrad. Therefore, the FOV/Solar disc = 0.52/8.7 = 1/16.7. If I understand the text 1/8 should be 1/16, if not some clarification is needed.

Page 17. Line 21: states the TEC-detector has a "narrower" bandwidth. Fig. 8b caption states the bandwidth is 100MHz "double band bandwidth". Is the single bandwidth 0-50MHz or 50-100 MHz in both sidebands? Additional text in the figure would be helpful.

Table 1: A bit confusing. Define the Quantities below the table, e.g., what is - info content H[bit], DFS dx [-] etc. ... Also - error prop. = retrieved error??, Statistical [$\mu\pm\sigma$] = ??. A list/explanation below the table would help and limit searching in the full text.

Figures:

Fig. 2: For the size figure printed some label font inside figures is too small and difficult to read on paper. Some colors do not show well making reading even worse.

[Figure]

Fig. 3: Font size in Fig. (a) is small for the size of the figure as presented. Fig.(a) is an important figure for the instrument discussion. Fig. 3(b) label font is unreadable in paper version. To make it useful either the figure has to be made significantly larger or the size of font increased.

Fif. 5: The word "modulation" here, in other figures and in the text was confusing to me. "Change" would be better. I believe I am right that the direction of the "scan" is from right to left on the page. If so, an arrow indicating that on the figure would be helpful.

Fig. 7: Fig. 7(a) needs more explanation in the caption. Why the SNR bandwidths labeled 30MHz and 300MHz not 60MHz and 600MHz as on the left for dates observed? What does the caption in Fig. (b) mean? A bit more text may help. Fig. 7(c) I assume the black x's are data points with wide scatter in same day measurements. Some discussion in caption would help.

Fig. 8. See above in text.

Fig. 9: (a) caption needs more description. As in Fig. 2 comment the font size is too small to read on paper. Some colors make it even harder.

---

## Author Comment (AC2) · 24 Aug 2016

Dear Dr Hase,

The holiday season delayed our reply to Dr Feist. We believe that this reply, now published, may address some of your concerns. Your comments regarding the assessment of the accuracy of XCO2 are very well aligned with our own roadmap for the next steps in the project, as outlined in the conclusion. We would like to reiterate that the primary focus of the manuscript is the description and performance assessment of the instrumental part of the measurement system. As such, only the analysis of instrumental error propagation has been so far performed. Since the instrumental performance is shown to be good ($\sim$ few times below the ultimate shot noise limit), the next step in the

[Figure]

maturation of the technique consists of addressing the issues you raised, in relation to uncertainty propagation of the geophysical inputs, to provide a full error budget on XCO2 measurements. To this aim, we have ~14 months of recorded data to be analysed. This is a significant amount of effort related to the development of our retrieval software (still currently in its infancy), in addition to cross-validation against well established, mature, instrumentation, and funding has been sought to deliver this follow-on work.

Comment 1 on the passive solar tracker and field deployability

In our manuscript, we have not given a figure quantifying the overall precision or accuracy of the solar tracking system assembly once deployed, since no solar disk imager was installed to allow us to do so. The various figures we give correspond to hardware specifications of the individual components (motorized stages), published precision estimates for the Sun position algorithm, modelled estimates for displacements over the course of a single acquisition, and lab-based pointing precision assessments of the assembled tracker, after mirror alignment but before deployment. Nevertheless, we found the system to qualitatively perform largely as anticipated, in as much as few adjustments are generally necessary over the time span of several hours of continued operation to keep pointing at the Sun. Of course, this is not a quantification of pointing accuracy at angles smaller than the arc subtended by the Sun disc, which is needed for retrieval error assessment. It is also not currently sufficient for autonomous and unsupervised operation. The presented research breadboard system is severely constrained by the building layout. The implementation of a solar disk imager was not practical (the distance between solar tracker and instrument is too long). We are aware that quantitative pointing accuracy data are needed to carry out the error budget already mentioned above. The field deployable version of the system currently being engineered will integrate a solar disk imager and the corresponding pointing analysis software with two objectives: 1) quantitative assessment of a passive pointing mechanism, and if required 2) active feedback for automated correction. Detailed assessment

of the tracking performance of such a system will follow once available, together with considerations about repeatability and campaign implications. As mentioned in the manuscript, the algorithm at the core of the tracker's control software can easily be up-graded as necessary, e.g. to include refraction corrections for high-latitude operation, or a slow component of control feedback.

Comment 2 on the degrees of freedom (DFS)

We have clarified the DFS in our response to comment 3 of SC1. We ought to clarify this more explicitly in the manuscript by mentioning the partial DFS of the individual retrieved quantities, making up the overall DFS. Retrieval sensitivity to spectroscopic parameters relates to bias analysis and geophysical input uncertainties. For all the input parameters, sensitivity matrices (D matrix in Rodgers terminology) shall provide the quantitative assessment; and spectroscopic parameters will be included. One of the advantages of using a narrow spectral window and a single transition is that spectroscopic parameters uncertainty propagation is more straightforward to analyse and control. Currently we haven't touched upon this particular task, which is part of the follow-on error budget. For the preliminary retrieval shown, data are taken directly from HITRAN 2012, and the specific parameters, as reported in HITRAN, are

- for $CO_2$: line position: 952.880849 +/- <0.00001 cm-1 (1) intensity: 1.900e-23 cm-1/molec.cm-2 +/- >20% (2) air-broadened half-width: 0.0793 cm-1/atm +/- <1% (3) self-broadened half-width: 0.109 cm-1/atm +/- <1% (4) temperature-dependence for air-broadening: 0.71 +/- <1% (5) air pressure-induced line shift: -0.002100 +/- <0.00001 cm-1/atm (6)

- for $H_2O$: line position: 953.367430 +/- <0.0001 cm-1 (7) intensity: 4.801e-24 cm-1/molec.cm-2 +/- 5-10% (8) air-broadened half-width: 0.0406 cm-1/atm +/- 5-10% (9) self-broadened half-width: 0.235 cm-1/atm +/- 2-5% (10) temperature-dependence for air-broadening: 0.39 +/- 10-20% (11) air pressure-induced line shift: -0.004630 cm-1/atm +/- <0.001 cm-1 (12)

So clearly there errors can be large and dominant as you suggest. Spectroscopic parameters accuracy will have to be significantly improved for the chose transitions. With an under-constrained problem, we agree that spurious profile oscillations are highly likely, and we have been facing such as well. Even before thinking of long-term measurement of real data, work on improved a priori constraints and vertical grid optimization have to be conducted and evaluated.

Comment 3 on errors associated with imperfectly known temperature profiles

We agree that the temperature sensitivity of the measurement needs to be looked into and analysed. We refer to our reply to comment 2 as well as the introductory paragraph to this reply. At this stage, the OSS has been developed to assess instrumental errors only (p. 7 l. 13) and determines how close to a perfect LHR the research system operates. We do not claim operational readiness but rather a first step in maturing the approach towards this goal. Biases due to errors in geophysical inputs, including the temperature profiles, are not included on purpose, and are subject to follow-on work (p. 19 l. 25). Our plan consists of using temperature (and pressure) data generated as space- and time-interpolated profiles from operational NWP analyses, with associated uncertainties (which are typically hard to quantify and/or obtain). These uncertainties will need propagating into the uncertainties of the retrieved $CO_2$ product(s). Whilst we agree that temperature variations in the PBL are frequently well beyond 10K, the uncertainty associated with interpolated 3h or 6h analyses (or higher-resolved forecast data) can be assumed to be much less, in particular throughout the troposphere. The sensitivity analysis should also provide valuable information informing on an optimal measurement protocol to be used.

Comment 4 on cross-comparison and validation

Further to the above, we re-emphasize that validation forms an integral part of follow-on work. We have collected data over a full year and (as far is possible in often cloudy Britain) over diurnal cycles. Corresponding assessments of cyclic variability will be

included in that work. Our research instrument remains available for future tests and inter-comparisons, particularly with our on-site FTS that is currently being upgraded towards TCCON requirements. The GOSAT monthly mean data point comparison was included as a coarse independent cross-check (sanity check) rather than as a comprehensive validation argument. We could emphasize this further in the manuscript if it is felt to be not clear enough. The use of modelled data will also be considered in follow-on work, either for comparison, as you suggest, but also in order to evolve our retrieval software with more rigorous a priori specification.

Spectroscopy references (taken from Hitran 2012): (1) 18. S.A. Tashkun and V.I. Perevalov. Private communication (2012). Line positions calculated from empirical energy levels, derived using RITZ approach (2) 27. S.A. Tashkun and V.I. Perevalov. Private communication (2012). Intensities calculated using the effective dipole moment function (3) 8. V. Malathy Devi, D. Chris Benner, M.A .H. Smith, L.R. Brown, and M. Dulick, "Multispectrum analysis of pressure broadening and pressure shift coefficients in the 12C16O2 and 13C16O2 laser bands," JQSRT 76, 411-434 (2003) (4) 7. Same as (3) (5) 5. R.R. Gamache and J. Lamouroux, "Predicting accurate line shape parameters for CO2 transitions," JQSRT 130, 158-171 (2013) (6) 10. Same as (3) (7) 30. R.A. Toth, "Linelist of water vapor parameters from 500 to 8000 cm-1," see http://mark4sun.jpl.nasa.gov/h2o.html (8) 18. Same as (7) (9) 24. Same as (7) (10) 1. For perpendicular bands derived from R.A. Toth, L.R. Brown, and C. Plymate, "Self-broadened widths and frequency shifts of water vapor lines between 590 and 2400 cm-1," JQSRT 59, 529-562 (1998), for parallel bands from R.A. Toth, JPL, unpublished (11) 22. n averaged as a function of J" for J"=0 to 13 from the work of M. Birk and G.Wagner, "Temperature-dependent air-broadening of water in the1250–1750 cm-1 range, JQSRT 113, 889-928 (2012) (12) 6. Same as (7)

---

## Short Comment (SC2) · 29 Aug 2016

Dear Dr. Weidmann,

thank you for your detailed reply. I am especially glad that you were able to clarify the high number of degrees of freedom (DOF) in your retrieval. Your explanation is fine. I was under the wrong impression that all the DOFs were related to the CO2 profile. This does not seem to be the case and the resulting number of CO2-DOFs is much closer to what I would have expected. This should be clarified in the manuscript as both myself as well as the editor did not interpret the text in the way you explained it in your comment.

About air mass: relying on external atmospheric profiles is risky. You should carefully estimate the error contribution from this approach on your retrieval scheme. The TC-CON experience is that even a high-accuracy (better than 0.1 hPa) surface pressure measurement on site - which is not a simple task itself - is not as good as actually measuring the O2 column.

I am content with your other replies.

Kind regards

Dietrich Feist

---

## Referee Comment (RC2) · Anonymous Referee #3 · 10 Oct 2016

The paper by Hoffman et al. presents a laser heterodyne spectrometer that enables measurements of the column-average mole fraction of carbon dioxide (XCO2) and, potentially, its vertical profile. The technique is carefully evaluated through retrieval simulations and through spectrometer testing in the lab. An appealing advantage of the proposed technique could be the vertical profiling aspect. The weak point of the study is the imbalance between lengthy discussion of theoretical and in-lab performance compared to a rather short section on the atmospheric deployment.

In fact, while more than a year of atmospheric data seem to be available, only one day of XCO2 is discussed. Why is that? Is there unforeseen real-world problems? I would assume that the real-world vertical profiling capability, for example, suffers from

real-world spectroscopic line parameter and line-shape uncertainties. I would urge the authors to discuss such real-world issues in more depths.

Nevertheless, the paper certainly deserves publication in AMT, since a relatively new atmospheric measurement technique is presented, its technical aspects are well documented and, the paper does include a first attempt on atmospheric deployment. I recommend taking into account the points below:

P8,L30: Make clear that 8 DFS refers to the entire state vector, not the CO2 profile part. Mention the number of DFS for the CO2 vertical profile. It would also be essential to describe where the height information comes from i.e. pressure/temperature dependence of the absorption lines. It might be worthwhile mentioning that spectroscopic parameter or lineshape errors would be highly detrimental.

Table 1, Figure 1, P8 first paragraph: The reason that the H2O retrievals are off the truth is that the a priori state vector is not equal to the truth and that the averaging kernel is not the identity matrix, right? So, actually, this is just a spurious smoothing effect driven by the (accidental) choice of prior and true H2O profile. I would think that this is of minor relevance for performance evaluation of a new instrument concept and it might distract the reader from the relevant parts.

Section 3.2: I am not convinced that a passive solar tracker is the preferred system for an application that uses an extremely narrow field of view (1/8 of the sun diameter) thus heavily relying on exact tracking of the solar disc center. The upper limits discussed would make a large contribution to or even exceed the tolerable error budget for XCO2. If cloud occurrence and subsequent loss of the solar tracking was the only concern, one could think about an active system that goes into passive mode once the intensity on the detector decreases.

P10, L30: How would the proposed system enable a significantly higher number of observations? Integration times of 90 s are not particularly fast. The FTS (Bruker HR125, EM27/SUN) typically used for ground-based XCO2 measurements can be at
least as fast (and probably still provide better SNR than the proposed LHR).

Section 5.2: If I understand section 5.2 correctly, only the XCO2 error bars as estimated by the retrieval are discussed. Given that no real validation against independent data is possible (1 GOSAT overpass is essentially insignificant), the authors could discuss how the estimated precision (1.9 ppm) compares to the observed data scatter e.g. estimated through the standard deviation of all soundings with respect to the moving average. One might assume with some justification that, on the timescale of 15 min, XCO2 is constant.